

# Seasonal dynamics of Totten Ice Shelf controlled by sea ice buttressing

Chad A. Greene[1], Duncan A. Young[1], David E. Gwyther[2], Benjamin K. Galton-Fenzi[3,4], and Donald D. Blankenship[1]

[1]Institute for Geophysics, Jackson School of Geosciences, University of Texas at Austin, Austin, Texas, USA.
[2]Institute for Marine and Antarctic Studies, University of Tasmania, Hobart, Tasmania, Australia
[3]Australian Antarctic Division, Kingston, Tasmania 7050, Australia
[4]Antarctic Climate & Ecosystems Cooperative Research Centre, University of Tasmania, Hobart, Tasmania 7001, Australia

**Correspondence:** Chad A. Greene (chad@chadagreene.com)

**Abstract.** Previous studies of Totten Ice Shelf have employed surface velocity measurements to estimate its mass balance and understand its sensitivities to interannual changes in climate forcing. However, displacement measurements acquired over timescales of days to weeks may not accurately characterize long-term flow rates where ice velocity fluctuates with the seasons. Quantifying annual mass budgets or analyzing interannual changes in ice velocity requires knowing when and where

observations of glacier velocity could be aliased by subannual variability. Here, we analyze 16 years of velocity data for Totten Ice Shelf, which we generate at subannual resolution by applying feature tracking algorithms to several hundred satellite image pairs. We identify a seasonal cycle characterized by a spring to autumn speedup of more than 100 m yr$^{-1}$ close to the ice front. The amplitude of the seasonal cycle diminishes with distance from the open ocean, suggesting the presence of a resistive backstress at the ice front that is strongest in winter. Springtime acceleration precedes summer surface melt and is not attributable to

thinning from basal melt. We attribute the onset of ice shelf acceleration each spring to the loss of buttressing from the breakup of seasonal landfast sea ice.

## 1 Introduction

Totten Glacier in East Antarctica drains the Aurora Subglacial Basin, which is grounded well below sea level (Young et al.,

2011; Roberts et al., 2011) and contains enough ice to raise the global sea level by at least 3.5 m (Greenbaum et al., 2015). Short-term observations have identified Totten Glacier and its ice shelf (TIS) as thinning rapidly (Pritchard et al., 2009, 2012) and losing mass (Chen et al., 2009), but longer-term observations paint a more complex picture of interannual variability (Paolo et al., 2015; Li et al., 2016; Roberts et al., 2017; Greene et al., 2017a). The current best estimates of Totten Glacier and TIS mass budgets have been calculated using a mosaic of surface velocity measurements collected at different times throughout the

year (Rignot et al., 2013); however, such estimates have been built on an unconfirmed assumption that ice velocity does not





vary on subannual timescales. Where glacier flow varies throughout the year, it is possible that velocity measurements collected over short time intervals may lead to to inaccurate estimates of annual mass balance or incorrect interpretation of interannual changes in velocity. Furthermore, most common methods of ice velocity measurement, such as satellite image feature tracking or in-situ GPS measurements taken over the course of a field season, are strongly biased toward summer acquisition and may
not accurately represent winter ice dynamics. Wherever seasonal velocity variability exists, it is important to consider how ice velocity is measured, and how the measurements can be interpreted.

Seasonal variations in glacier velocity have been observed in Greenland and Antarctica (e.g., Joughin et al., 2008; Nakamura et al., 2010; Moon et al., 2014; Zhou et al., 2014; Fahnestock et al., 2016), and have been attributed to a number of different mechanisms. On grounded ice, surface meltwater can drain into crevasses or moulins, make its way to the bed, pressurize inef-
ficient subglacial hydraulic systems, and allow the glacier to speed up until pressure is reduced (Sohn et al., 1998; Bartholomew et al., 2010; Moon et al., 2014). On floating ice, surface meltwater may also influence ice shelf velocity by percolating through and weakening the ice shelf shear margins (Liu and Miller, 1979; Vaughan and Doake, 1996). Observations have shown correspondence between seasonal advance and retreat of marine-terminating glaciers and the presence of ice melange at the glacier terminus (Howat et al., 2010; Cassotto et al., 2015; Moon et al., 2015). The exact mechanisms by which ice melange can affect
glacier dynamics are poorly understood, but modeling studies have shown that the back stress provided by sea ice can prevent calved icebergs from rotating away from the ice front (Amundson et al., 2010), and in some cases can shut down calving entirely (Robel, 2017) causing an appreciable effect on glacier velocity (Todd and Christoffersen, 2014; Krug et al., 2015). For example, the buttressing strength of ice melange at Store Glacier in Greenland has been estimated at 30–60 kPa, which is an order of magnitude below the driving stress of the glacier, but is sufficient to cause observable subannual changes in glacier
velocity up to 16 km from the ice front (Walter et al., 2012; Todd and Christoffersen, 2014).

In Antarctica, marine ice is known to strengthen the Brunt and Stancomb-Willis ice shelf system (Hulbe et al., 2005), and an ice shelf acceleration event observed there in the 1970s has been attributed to a reduction in stiffness of the ice melange that connects the two ice shelves (Khazendar et al., 2009). Similarly, multi-year landfast sea ice is strongly mechanically coupled to Mertz Glacier Tongue (Massom et al., 2010) and may have delayed a major calving event that occurred there in 2010 (Massom
et al., 2015). Closer to TIS, two recent major calving events in Porpoise Bay (76°S,128°E) were attributed to the breakup of landfast sea ice at the ice shelf termini (Miles et al., 2017). At TIS, long-term changes in calving front position have been reported with a possible connection to local sea ice processes (Miles et al., 2016), but corresponding links to glacier dynamics have not previously been investigated. To our knowledge, there have been no reports of seasonal variability of TIS or any of the mechanisms that may drive TIS variability at subannual timescales. In this paper we find seasonal variability in two
independent ice velocity datasets and we consider the potential roles of surface melt water, ice shelf basal melt, and sea ice buttressing, in influencing the flow of TIS at subannual timescales.





## 2 Surface velocity observations

We analyzed surface velocity time series using feature tracking algorithms applied to Landsat 8 and MODIS (MODerate-resolution Imaging Spectroradiometer) images. Each image dataset was processed separately, using different feature tracking programs, and the resulting time series represent two independent measures of TIS velocity. The 15 m resolution of Landsat 8 permits precise displacement measurements over short time intervals, but the relatively brief four-year Landsat 8 record and limited number of cloud-free images inhibits our ability to separate interannual velocity changes from seasonal variability. The MODIS record contains many cloud-free images per year from 2001 to present; however, the 250 m spatial resolution of MODIS images limits measurement precision where ice displacements are small between images. Thus, the two image datasets each offer incomplete, but complementary insights into the seasonal dynamics of TIS. Processing methods for each dataset are described below.

### 2.1 GoLIVE (Landsat 8) velocities

We used the Global Land Ice Velocity Extraction from Landsat 8 (GoLIVE) dataset (Scambos et al., 2016; Fahnestock et al., 2016), which is processed at 600 m resolution for most of Antarctica. We analyzed the high-confidence `vx_masked` and `vy_masked` velocity fields from late 2013 to early 2018 and limited the dataset to 143 image pairs separated by $16 \leq dt \leq 112$ days. Many of the image pairs overlap in time, providing several redundant, semi-independent velocity measurements, particularly throughout the summer months when each image may contribute to multiple image pairs.

To understand the spatial pattern of TIS seasonality, we developed characteristic velocity maps for spring and autumn separately. Spring velocity was taken as the mean of 76 velocity measurements whose image pairs were obtained between June 16 and December 15. Autumn velocity was taken as the mean of 67 velocity fields from images obtained during the remainder of the year. We discard all pixels where the mean ice speed is less than 250 m yr$^{-1}$. We also discard all pixels containing fewer than 10 high-confidence spring or autumn velocity measurements. The resulting difference between spring and autumn velocities is shown in Fig. 1.

The terminal ∼50 km of TIS accelerates each year from spring to autumn, then slows throughout the winter. Seasonality is strongest close to the glacier terminus and decays with distance from the open ocean. The relatively featureless nature of the inner TIS surface limits the number of high-confidence matches in that region of the ice shelf, but the available measurements indicate minimal seasonality upstream of the mid-shelf ice rumple identified by InSAR (Mouginot et al., 2017b). The grounded ice of the eastern tributary accelerates slightly throughout the summer, while some grounded ice of the western tributary exhibits a weak slowdown.

To assess the timing of the annual TIS acceleration, we generate a velocity time series for a region of TIS near the terminus shown in Fig. 1. We populate a velocity time series from the means of all GoLIVE velocity measurements within 5 km to 10 km from the ice front, considering only pixels with a mean velocity exceeding 1700 m yr$^{-1}$. The resulting TIS velocity time series is shown in Fig. 2.




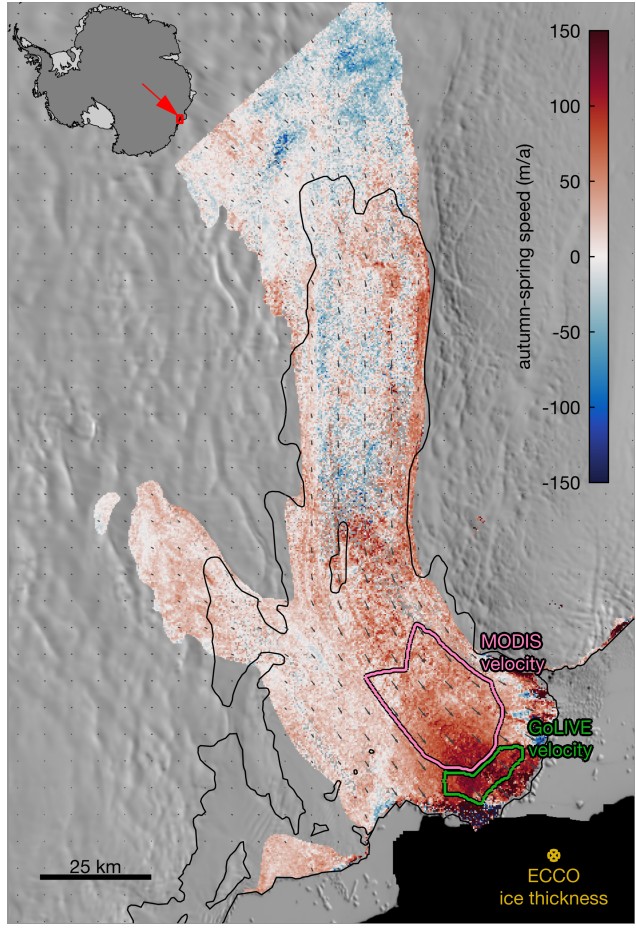

**Figure 1. Summer ice shelf acceleration.** Toward the ice front, autumn velocity exceeds spring velocity by more than 100 m yr$^{-1}$. This image shows the difference between the means of 76 spring and 67 autumn GoLIVE velocity fields. Dark green vectors indicate the mean velocity, supplemented by MEaSUREs InSAR-derived velocity (Rignot et al., 2011) outside the range of Landsat path 102, row 107. Green and pink polygons indicate the bounds of velocity averaging for the velocity time series shown in Fig. 2 and Fig. 3, respectively. A gold marker shows the location of the ECCO sea ice thickness time series described in Sec. 5.

The short record and low temporal resolution of the GoLIVE dataset make it difficult to identify the exact timing of the onset of acceleration in any given year, but a linear trend fit to all available measurements indicates a typical acceleration of 0.8 m yr$^{-1}$ per day from late September to early April. Further investigation into the timing of accelerations each year requires a more complete time series of TIS velocity, which we generate from MODIS images.

5 **2.2 MODIS velocities**

A MODIS velocity time series was generated from 672 pairs of cloud-free MODIS band 2 images (Scambos et al., 2001, updated 2018; Greene and Blankenship, 2017) acquired between 2002 and 2018. Each image pair was separated by 92 to 182



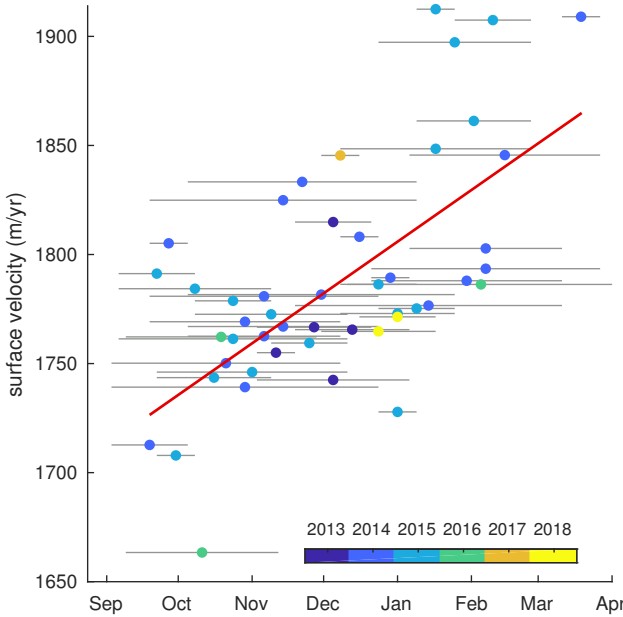

**Figure 2. Ice front acceleration from GoLIVE (Landsat 8).** The GoLIVE dataset contains many overlapping TIS velocity measurements captured between September and April of each year. The velocities here represent all displacements measured over $16 \leq dt \leq 112$ days (indicated by gray lines), averaged within the green polygon in Fig. 1. The red trend line is a linear least-squares fit to the observations, indicating a typical spring-to-fall acceleration of 0.8 m yr$^{-1}$ per day.

days and was processed at 250 m resolution using the ImGRAFT template matching software (Messerli and Grinsted, 2015) with Antarctic Mapping Tools for MATLAB (Greene et al., 2017b). Similar to the method described by Greene et al. (2017a), we used 2.5 km square templates with 4.0 km search boxes centered on locations predicted by InSAR-derived velocities (Rignot et al., 2017). To generate the MODIS velocity time series we averaged velocities from all pixels within 10 km to 30 km from

5 the ice front, bounded on each side by the glacier shear margins identified by Greene et al. (2017a). We discarded any image pairs for which fewer than 99% of the pixels within the polygon contained valid displacement measurements, resulting in 565 valid MODIS velocity measurements in the time series. The polygon used for the MODIS time series is shown in Fig. 1.

Despite having measurements from dozens of MODIS image pairs most years, subannual template matching applied to 250 m resolution MODIS images produces such noisy velocity estimates that the timing of springtime acceleration cannot

be accurately determined for any given year. Figure 3 shows the MODIS velocity time series overlaid on the mean seasonal cycle. Because no visible-band MODIS images are available during the dark winter months, no image pairs separated by 92 to 192 days are centered on any days in April, May, August, or September. However, 46 image pairs span the winter, providing velocity measurements centered on June and July.

We approximate the seasonal cycle of the TIS velocity as a sinusoid obtained by least squares fit to the 565 MODIS ve-

15 locity measurements. To minimize the influence of interannual variability, the one-year moving average was removed before



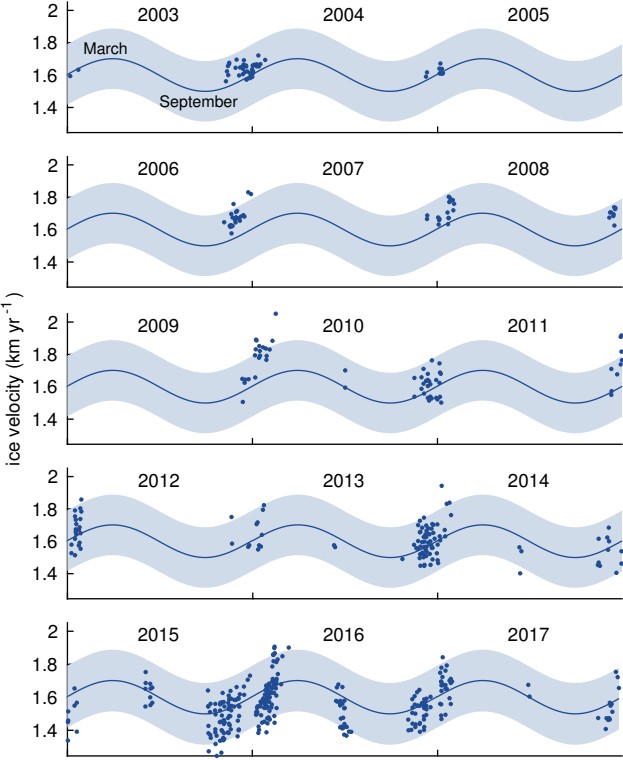

**Figure 3. Seasonal cycle of ice shelf velocity from MODIS.** TIS velocity measurements from 565 MODIS image pairs separated by 92 to 182 days, averaged within the pink polygon shown in Fig. 1. The average seasonal cycle is shown approximated as a sinusoid, with 95% confidence intervals shaded.

analyzing the seasonal cycle. The resulting best-fit sinusoid is characterized by a 1601 m yr$^{-1}$ mean velocity, an amplitude of 106 m yr$^{-1}$, a maximum velocity on March 21, and a minimum velocity on September 19. The sinusoid matches observations to a root-mean-square error of 93 m yr$^{-1}$.

## 3  Surface melt observations

5  Surface melt has been shown to affect the flow of grounded ice in Greenland when surface water drains through moulins or crevasses to the bed, where it alters basal water pressure and allows the overlying ice to accelerate (Zwally et al., 2002; Schoof, 2010; Bartholomew et al., 2010; Andrews et al., 2014). The seasonal velocity anomalies we observe at TIS are strongest near the floating ice front, so it is unlikely that the seasonal variability of TIS velocity is driven by subglacial hydrology on nearby grounded ice. However, the presence of englacial liquid water can weaken ice (Liu and Miller, 1979), and it is plausible that

10  surface melt at TIS could percolate into the ice, weaken shear margins, and allow TIS to speed up as a result of reduced buttressing.



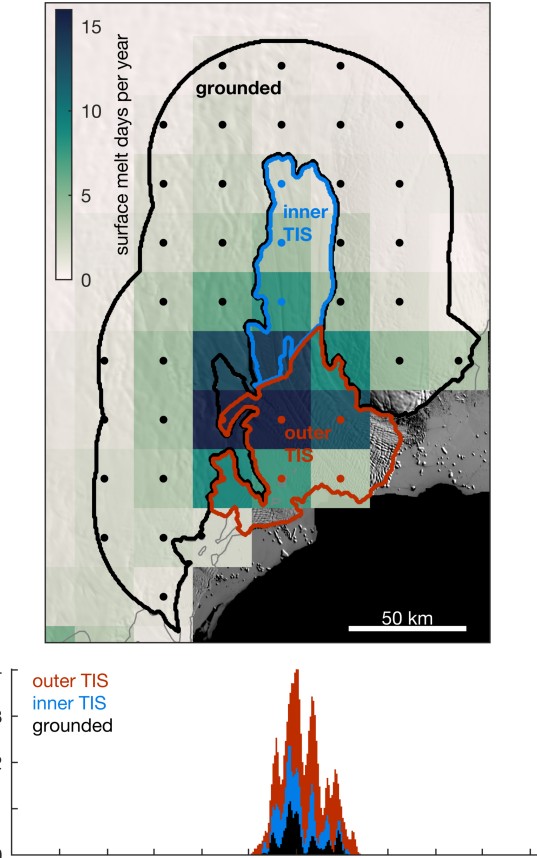

**Figure 4. Surface melt observations, 2000–2017.** Spatial distribution of mean annual surface melt and daily probability of surface melt in each subdomain. Low-elevation areas near the coast experience more days of surface melt than high-elevation grounded ice, but the timing of surface melt is similar throughout the domain. Data are from Picard and Fily (2006).

To assess the possible link between surface melt and TIS velocity anomalies, we used daily observations of surface melt from passive microwave radiometers (SMMR and SSM/I) gridded to 25 km resolution (Picard and Fily, 2006). We limited the period of analysis to 2000 through 2017 to roughly coincide with available MODIS image data. Figure 4 shows the spatial distribution of mean annual surface melt during this period. Using the masks developed by Mouginot et al. (2017b) with the Antarctic Mapping Tools for MATLAB `dist2mask` function (Greene et al., 2017b), we define three subdomains for surface melt analysis as

1. *outer TIS:* the floating portion of the ice shelf up to 50 km from the ice shelf front,

2. *inner TIS:* the floating portion of the ice shelf more than 50 km from the ice shelf front, and

3. *grounded:* all grounded ice within 50 km of the TIS grounding line.





Surface melt is most prevalent in the outer TIS, where in some locations surface melt is detected up to 16 days per year. Fewer surface melt days occur far from the ice front on the inner TIS, and surface melt is least common on the high-elevation grounded ice surrounding TIS. Figure 4 shows that although the number of annual surface melt days varies throughout the region, the timing of surface melt is roughly the same in all three subdomains, with the typical melt season lasting from

December to February. For the outer TIS, the onset of surface melt typically occurs on December 23 ($\pm 1\sigma = 12$ days), with the earliest summer melt recorded on December 6, 2006. The mean final day of surface melt occurs on January 23 ($\pm 1\sigma = 9$ days), but has been observed as late as February 11 in 2005.

## 4 Modelled ice shelf basal melt

On interannual timescales, TIS is known to accelerate in response to prolonged periods of elevated basal melt rates (Roberts

et al., 2017; Greene et al., 2017a), and a similar process has been observed at Pine Island Ice Shelf in West Antarctica (Christianson et al., 2016). For these laterally-bounded ice shelves restrained largely by shear stress at their margins, thinning reduces resistance to glacier flow and allows ice shelf acceleration.

To assess whether TIS may dynamically respond to basal melt anomalies at subannual timescales, we used the Regional Ocean Modeling System (ROMS; Shchepetkin and McWilliams, 2005) to simulate TIS-ocean interactions, then considered

the effects that subannual basal melt anomalies could have on TIS velocity. The model domain extended from 104.5°E–130°E in longitude and 60°S–68°S in latitude with a horizontal resolution of approximately 2 to 3 km. A terrain-following vertical coordinate provided enhanced resolution close to the seafloor and ice shelf interface. Modifications to the code allowed thermodynamic interaction between ocean and steady-state ice shelves, following Dinniman et al. (2003) and Galton-Fenzi et al. (2012). Seafloor topography was based on the RTOPO dataset (Timmermann et al., 2010), while cavity geometry was

inferred from ICESat-derived ice surface elevation above flotation and a constant 300 m thick offset along the central flow line. Between the central flow line and the grounding line, cavity bathymetry was linearly interpolated (see Gwyther et al., 2014, for details). The model lateral and surface boundaries were forced over the hindcast period 1992–2012. Lateral forcing was derived from the ECCO2 cube92 reanalysis solution (Menemenlis et al., 2008); surface forcing was ERA-interim wind stress (Dee et al., 2011), and heat and salt fluxes were derived from Special Sensor Microwave/Imager (SSM/I) algorithms for

sea-ice production (Tamura et al., 2016). The model was spun-up for 21 model years using 1992–2012 forcing. After spin-up, the 1992–2012 forcing was repeated and we analyzed the mean seasonal cycle of melt from the second run. The mean spatial distribution and temporal variability of basal melt are shown in Fig. 5.

The distribution of basal melt at TIS mimics observations of other ice shelves, with the highest melt rates focused near the deep grounding line (e.g., Dutrieux et al., 2013). Seasonal variability is most significant in the inner TIS, where mean melt

rates are highest, whereas the shallow ice base of the outer TIS experiences only a weak seasonal cycle superimposed on a low mean melt rate. Everywhere beneath the ice shelf, basal melt rate reaches a maximum in autumn and a minimum in the spring.

Ice shelf thinning tends to reduce buttressing and allow ice shelf acceleration. Using a simple model developed by Greene et al. (2017a) (adapted from Joughin et al., 2004) to estimate velocity anomalies resulting from seasonal changes in ice thick-





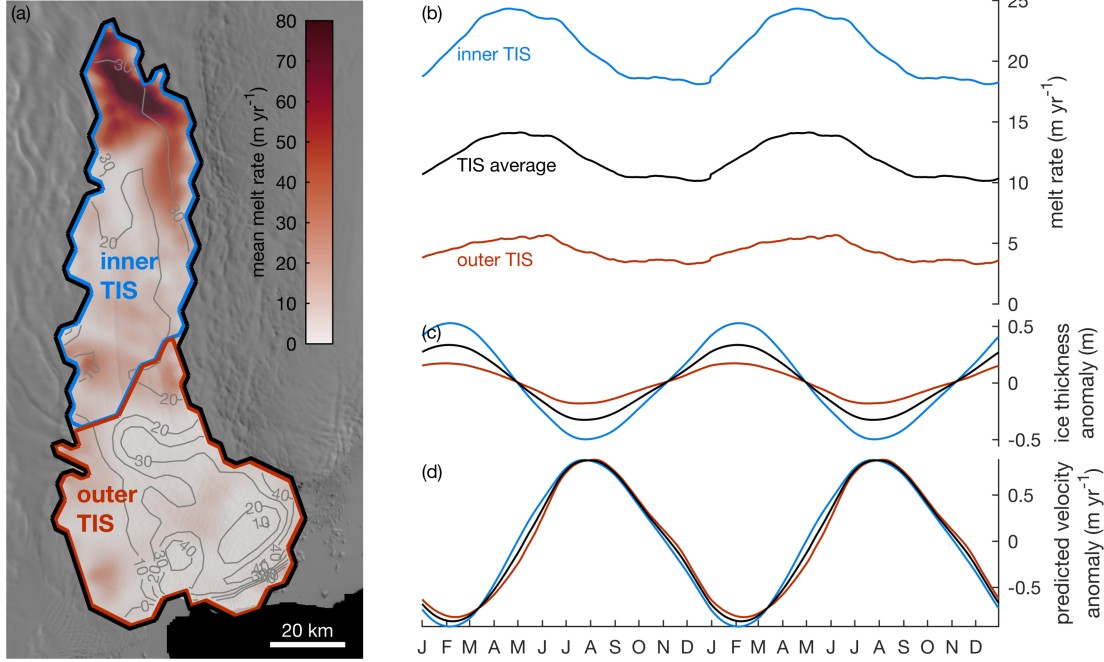

**Figure 5. Modeled basal melt. (a)** Mean melt rate distribution shows melt focused where the ice shelf base is deepest, exceeding $80\,\mathrm{m\,yr^{-1}}$ near the grounding line of the inner TIS. Gray contours show melt rate lag times in days relative to anomalies at the ice front, indicating melt anomalies propagate in a clockwise fashion around the cavity. **(b)** Two years of 1992–2012 climatological melt rates averaged within the subdomains in **(a)**. **(c)** Ice thickness anomalies corresponding to the time integral of melt rate anomalies. **(d)** Ice velocity anomalies expected to result from seasonal variations in ice shelf thickness.

ness, we find that on subannual timescales, basal melt anomalies should only affect TIS velocity on the order of $1\,\mathrm{m\,yr^{-1}}$ (Fig. 5). Note that velocity predictions are negatively correlated with ice thickness, which is calculated from the time integral of basal melt rate anomalies. Accordingly, velocity maxima related to basal melt do not correspond directly to basal melt rate maxima, but should occur at the end of the high-melt season in July, when ice thickness reaches a minimum.

5   Small perturbations in ice shelf thickness have the greatest influence on ice shelf buttressing where the ice shelf is thin. However, the thick ice of the inner TIS experiences much more seasonal melt variability than the thin ice of the outer TIS, so it is somewhat by coincidence that the large ($>8\,\mathrm{m\,yr^{-1}}$) increase in melt rate from spring to autumn beneath the thick ($>2000\,\mathrm{m}$) ice of the inner TIS, affects local ice velocity to approximately the same degree as the much smaller ($\sim 3\,\mathrm{m\,yr^{-1}}$) seasonal melt rate variability in the outer TIS, where ice is much thinner (Fig. 5).

10   The model we use to estimate melt-induced velocity anomalies assumes TIS velocity is limited only by lateral shear stress at the ice shelf margins and velocity anomalies are purely a function of local ice thickness. These assumptions vastly oversimplify the complex stress regime of the TIS, but are used to obtain an order-of-magnitude approximation of how the TIS should respond to seasonal variability of ice thickness driven by basal melt. From this simple model it is clear that the $<1\,\mathrm{m\,yr^{-1}}$





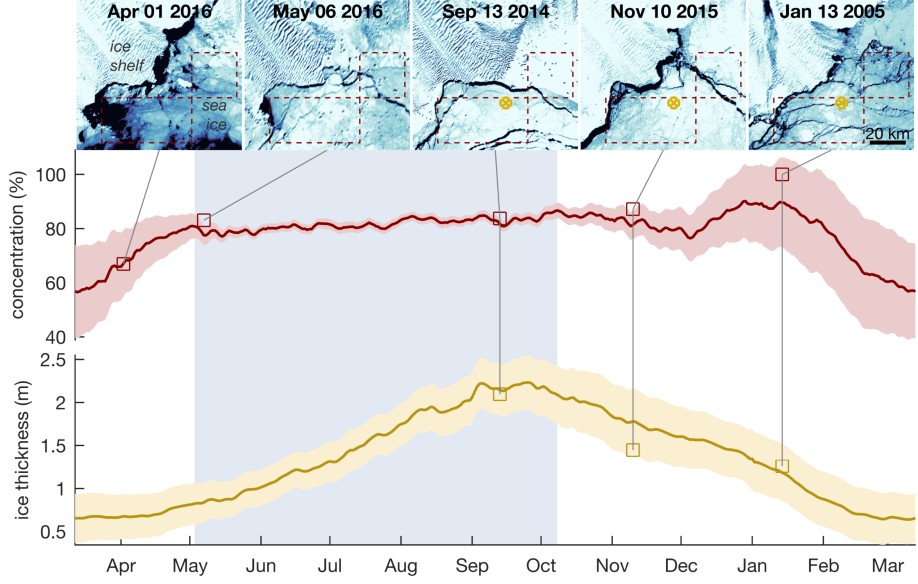

**Figure 6. Seasonal cycle of sea ice at the TIS front.** MODIS visual band imagery (Scambos et al., 2001, updated 2018), remotely sensed ice concentration (Cavalieri et al., 1996), and ECCO v4-r3 effective ice thickness (Fukumori et al., 2017) reveal a seasonal cycle of sea ice growth and decay beginning around March 12 each year, when sea ice concentration is at a minimum. Ice concentration and thickness time series are shown with corresponding shading indicating daily values of $\pm1\sigma$. Light blue background shading indicates the presence of fast ice at the TIS front, from about May 3 to October 8. During this winter period, remotely sensed ice concentration values remain relatively constant despite continued growth of sea ice. Average ice thickness steadily declines throughout the summer, while thin, unconsolidated sea ice often temporarily fills the area and is detected by remote sensors. Five example MODIS images (Scambos et al., 2001, updated 2018) are shown for context, with dashed quadrangles indicating the region of ice concentration averaging and a gold marker denotes the location of the ECCO ice thickness time series.

variability expected to result from seasonal basal melt anomalies cannot explain the observed $>100$ m yr$^{-1}$ seasonal variability of TIS velocity. Moreover, holding other factors are constant, the seasonal cycle of basal melt produces an ice shelf that grows throughout the summer and reaches a maximum thickness in February. Accordingly, basal melt anomalies should result in a summer slowdown and a velocity minimum in February, when observations show TIS nearing its velocity maximum. Thus, it

5    is unlikely that the seasonal cycle of basal melt could explain the observed pattern of spring-to-fall acceleration of TIS.

## 5    Sea ice concentration and thickness

To assess whether the presence of sea ice may influence the flow of TIS, we analyzed observational data from microwave, thermal, and visual band satellite sensors, along with model data of ice thickness near the TIS front. We used daily observations of sea ice concentration (Cavalieri et al., 1996) and generated a time series given by the mean of three 25 km grid cells located

10   close to the TIS front (shown in Fig. 6). In addition to ice concentration observations, we also analyzed daily effective sea





ice thickness from ECCO v4-r3 for the period 2000–2015 (Fukumori et al., 2017). We focused on the time series of sea ice thickness for the grid cell centered on (66.47°S,116.50°E), indicated by gold markers in Figs. 1 and 6. To fully understand the spatial and temporal variability of sea ice, we also inspected 315 cloud-free MODIS visual (band 2) and 164 thermal (band 32) images acquired throughout the year by the Aqua and Terra platforms between 2000 and 2017 (Scambos et al., 2001, updated

2018; Greene and Blankenship, 2017).

Figure 6 shows the seasonal cycle of sea ice growth and decay. The minimum ice concentration typically occurs at the TIS front in mid March, followed by increasing ice concentration throughout autumn as air temperatures decline (Dee et al., 2011). Inspection of visual and thermal band images reveals that sea ice consolidates and fastens to the western TIS in early to mid May. The rigid connection of landfast ice to the TIS front holds throughout the winter, with the exception of a small polynya

abutting Law Dome that briefly opened in July 2016 (see Alley et al., 2016). Regardless of polynya activity, the majority of landfast ice remains connected to the TIS front, and each year the landfast connection breaks in October or early November, followed by a visible reduction in sea ice cover that occurs throughout November. In some years, sea ice concentration continues to decline throughout the summer, but more commonly, the region temporarily fills with unconsolidated ice, causing sea ice concentration to peak in January (Greene, 2017). From January to March, sea ice melts or is exported away from the TIS

front until concentration reaches a minimum in mid March, then the cycle repeats. Ice thickness data are more well behaved, generally waxing and waning monotonically between a minimum in late March and maximum in late September. In this way, ice thickness follows the broader climatology reported by Fraser et al. (2012), who found that landfast ice between 90°E and 160°E grows from a minimum extent in March to a maximum in late September or early October.

We suspect that sea ice concentration is a poor measure of ice strength, because the simple fraction of a grid cell's surface

area covered by ice offers no indication of ice thickness or level of consolidation. This is seen not only in the summer melt season during which sea ice concentration often increases, but also in winter, when ice concentration observations remain constant while the ice grows steadily thicker. We posit that ice thickness is a better proxy for ice strength because the ECCO v4-r3 model was indirectly constrained by observations of sea ice concentration, but also accounts for winter sea ice growth.

# 6 Discussion

## 25 6.1 Causes of seasonal variability

Studies of floating and marine-terminating glaciers around the world have found a diverse set of causes of seasonal velocity variability, suggesting that local phenomena control glacier flow on subannual timescales, and there is no single dominating global cause of seasonal variability. In some regions of Greenland, neighboring glaciers behave differently based on the mechanisms that control them, and those mechanisms can change for a given glacier throughout the year (Howat et al., 2010; Moon

et al., 2014).

On grounded ice, seasonal velocity variability often results from surface water draining to the bed, where it can temporarily pressurize an inefficient hydrological system, allowing the overlying ice to accelerate until an efficient drainage system forms or the water otherwise evacuates (Zwally et al., 2002; Parizek and Alley, 2004; Bartholomew et al., 2010). At Totten Glacier,





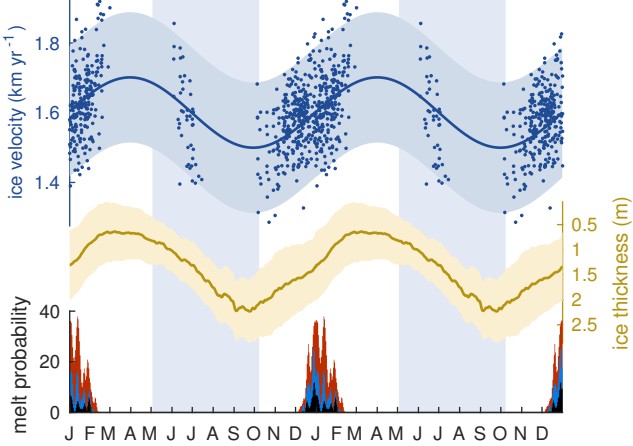

**Figure 7. Causes of springtime acceleration.** Time series of MODIS-derived ice velocity $\pm 2\sigma$ shaded, ECCO v4-r3 ice thickness$\pm 2\sigma$ shaded, and melt probability repeated from Figs. 3, 4, and 6. Note the inverted axis of the ice thickness time series. Springtime acceleration near the TIS front begins with the breakup of landfast sea ice, and is possibly enhanced later in the year by shear margin weakening resulting from surface melt. Vertical shaded blue areas indicate typical times of landfast ice connection with TIS. Two years of the mean cycle are shown for visual continuitiy. Velocity anomalies predicted from basal melt are not shown here becuase the $\pm 1$ m yr$^{-1}$ amplitude would visually indiscernible at the observed scale of interannual velocity variability.

we detect very little seasonal velocity variability on grounded ice, and the onset of acceleration we observe on the floating ice shelf begins well before surface water is detected anywhere in the region (Fig. 7). We therefore rule out the possibility that surface melt is responsible for initiating TIS acceleration each year.

On multi-year timescales, basal melt can lead to ice shelf acceleration as thinning reduces the internal buttressing strength
of ice (Christianson et al., 2016; Greene et al., 2017a). However, neither the timing nor the amplitude of melt-induced thinning can account for the seasonal velocity variability we observe at TIS. We find a small seasonal cycle of ice shelf thickness due to variable basal melt throughout the year, but the corresponding velocity anomalies should be two orders of magnitude smaller than the observed velocity anomalies. Roberts et al. (2017) pointed out that at TIS, a mechanism exists that can amplify the effects of basal melt on ice velocity: Ice rumples in the middle of TIS may provide decreasing resistance to flow as the ice shelf
thins, so it is possible that the simple model of ice shelf buttressing we employ in Sec. 4 underestimates the effects of basal melt on TIS velocity. Nonetheless, we find that basal melt is still incapable of causing the observed seasonal velocity cycle because ice shelf thickness maxima occur each year nearly coincident in time with observed velocity maxima.

In the GoLIVE dataset and in MODIS-derived velocities, we find that the outer TIS accelerates each year between spring and autumn. The spatial pattern of the annual acceleration suggests that the flow of ice is governed by processes at the ice
front (Fig. 1), and timing of the acceleration implicates the annual breakup of landfast ice at the TIS front as an influencing factor. Figure 7 shows the relationship between sea ice thickness, surface melt, and TIS velocity. The temporal and spatial resolution of data available at TIS limit our ability to investigate the specific processes by which the presence of sea ice may

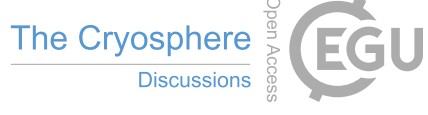



slow the flow of TIS, but similar studies elsewhere have found that backstress from ice melange (Walter et al., 2012; Todd and Christoffersen, 2014; Otero et al., 2017) can stabilize the ice front and reduce or entirely shut down calving over winter (Sohn et al., 1998; Reeh et al., 2001; Amundson et al., 2010; Moon et al., 2015; Robel, 2017), thus preserving internal stresses in the glacier and slowing its flow (Krug et al., 2015). The pattern of TIS acceleration we observe is similar to seasonal velocity

anomalies observed at other marine-terminating glaciers and ice shelves, where the annual breakup of sea ice causes velocity anomalies that are seen up to tens of kilometers from the glacier terminus (Nakamura et al., 2010; Walter et al., 2012; Zhou et al., 2014).

We find that the outer TIS accelerates each spring, likely in response to lost buttressing upon the breakup of rigid sea ice at the ice shelf terminus. The response we observe is consistent with other studies that have shown a seasonal pattern of ice

front calving and glacier acceleration in response to the disintegration of rigid sea ice caused by warm sea surface temperatures (Howat et al., 2010; Cassotto et al., 2015; Luckman et al., 2015). Ice front processes are likely responsible for the onset of TIS acceleration each spring, but we cannot rule out the possibility that other factors may influence the flow of TIS in other parts of the year. It is possible that onset of acceleration begins with the breakup of sea ice at the TIS front, but surface melt could play a role later in the summer or autumn, if water percolates into the ice and weakens the shear margins.

In Figs. 3 and 7 we approximate the seasonal velocity variability of TIS as a sinusoid. The seasonal flow of TIS is likely more complex, and the timing and magnitude of spring-to-autumn speedup presumably vary from year to year. Nonetheless, we have shown that TIS responds to local forcing on subannual timescales, the response is observable, and it correlates with the breakup of sea ice at the glacier terminus each spring.

### 6.2    Impacts of seasonal variability on measurements of long-term change

We find that TIS accelerates each year from spring to autumn, and this seasonal variability has the potential to contaminate estimates of annual mass flux and interannual variability. Most common methods of measuring ice velocity rely upon subannual displacement measurements to characterize annual ice flux (e.g. Mouginot et al., 2017a), but where ice velocity varies throughout the year, short-term measurements can be aliased by the natural seasonal cycle and provide an inaccurate measure of annual ice flux. The seasonal variability we observe at TIS suggests that measurements acquired in the spring likely underestimate,

and autumn measurements overestimate, the mean annual velocity of the ice shelf.

The most significant seasonal variability at TIS is found near the ice front, where spring and autumn velocities can differ by up to 10% percent. Although this represents a small modulation of the mean flow, it is on the order of interannual variability that has previously been attributed to interannual changes in ocean forcing, and the pattern of summer acceleration we show in Fig. 1 bears a notable resemblance to accelerations that have previously been reported as evidence of long-term change (Li

et al., 2016). Although direct investigations of interannual change are beyond the scope of this study, we can consider how seasonal variability may have influenced previous studies of TIS velocity.

Roberts et al. (2017) and Greene et al. (2017a) each found interannual changes in velocity by analyzing displacements between images separated by near-integer multiples of years. By this method, it is unlikely that they inadvertently captured subannual variability, unless the timing of acceleration events occurred out of sync with the calendar year. Such is likely the



case for the 2009 to 2010 acceleration observed by Li et al. (2016), who compared velocity measurements obtained between September and January of both years. Although the periods of observation were roughly the same in both years, the spring breakup of fast ice did not occur until after the start of observations in 2009, whereas the spring breakup was already underway when observations began in 2010, and the TIS had already begun to respond. The velocity difference between the 2009 and 2010 measurements shows acceleration focused at the ice shelf terminus, and this likely reflects a difference in timing of the seasonal cycle that may not be associated with any difference in mean annual velocities. The inconsistent timing of fast ice breakup each year suggests that assessments of interannual change made from short-term displacement measurements can be contaminated by seasonal effects, even if observations are taken at the same time each year.

Despite the seasonal variability we observe near the TIS front, mass balance of an ice sheet is more meaningfully measured at the grounding line, where ice begins to have an impact on sea level. Our results show little subannual velocity variability at the grounding line, thus supporting the grounding line flux estimates by Li et al. (2016).

### 6.3 Sea ice influence on ice sheet mass balance

The GoLIVE and MODIS velocity measurements show that TIS is sensitive to environmental forcing on subannual timescales, and its flow is primarily controlled by the presence and strength of sea ice at the TIS front. This finding warrants consideration of how changes in sea ice could affect the stability of the TIS and the long-term mass balance of the Aurora Subglacial Basin. Elsewhere in Antarctica, loss of multiyear landfast ice has led to major calving events and glacier acceleration (Khazendar et al., 2009; Miles et al., 2017; Aoki, 2017). The landfast ice we observe is not multiyear ice, and is thus unlikely to be associated with any catastrophic events at TIS in the near future. However, calving front processes can have far-reaching effects on glacier thickness and velocity (Nick et al., 2009), and it is possible that long-term changes in winter sea ice cover (Bracegirdle et al., 2008) could have integrated effects on TIS buttressing: The duration and thickness of sea ice cover each winter controls the total annual buttressing at the ice front, the annual flow of the ice shelf, and potentially the long-term mass balance of TIS and the Aurora Subglacial Basin.

### 7 Conclusions

We find that TIS has a characteristic seasonal velocity profile, which could lead to inaccurate estimates of the annual mass balance of TIS, and may have aliased some previous measurements of interannual variability. Annual ice velocity maps are now available covering most of Antarctica (Mouginot et al., 2017c), interpreting such datasets at TIS and elsewhere requires understanding where ice velocity varies seasonally and by how much. Our results provide context for how and where such velocity mosaics may be used to interpret interannual change at Totten Glacier.

Previous studies have investigated TIS velocity variability and have broadly concluded that interannual changes in ice velocity have been caused by sustained basal melt rate anomalies. Basal melt cannot explain the seasonal velocity variability we observe, because the seasonal amplitude of melt is too weak to produce enough thinning for an observable velocity response.



Furthermore, seasonal basal melt anomalies result in an ice shelf that is thinnest, weakest, and should flow the fastest in winter, when our observations show the TIS reaches its minimum velocity.

In accord with other studies of ice shelves and glaciers around Antarctica and Greenland, we find that the seasonal variability of TIS velocity is most closely linked to the presence of sea ice at the ice shelf front. Each spring when surface waters warm,
rigid landfast ice breaks its connection to the TIS front, the calving rate increases, and the TIS responds by accelerating by nearly 10% close the ice shelf terminus. Velocity anomalies are most significant over floating ice, and spring acceleration precedes surface melt each year, together suggesting that subglacial hydrology does not cause the seasonal cycle of TIS velocity we observe.

We find that winter sea ice is a primary contributor to the seasonal variability of the outer TIS velocity. If the future brings
long-term changes in the thickness or extent of winter sea ice, the integrated effects of changes in buttressing could manifest as long-term changes in the mass balance of TIS and the Aurora Subglacial Basin.

*Code and data availability.* GoLIVE data (Scambos et al., 2016; Fahnestock et al., 2016) is available at https://nsidc.org/data/NSIDC-0710. MODIS images (Scambos et al., 2001, updated 2018) used in this study were obtained from ftp://sidads.colorado.edu/pub/DATASETS/ICESHELVES. The ImGRAFT template matching software (Messerli and Grinsted, 2015)
is available at http://imgraft.glaciology.net. The `melting-1979-2017-v2.nc` surface melt data from Picard and Fily (2006) are available at http://pp.ige-grenoble.fr/pageperso/picardgh/melting/. ECCO v4-r3 sea ice effective thickness data can be found at ftp://ecco.jpl.nasa.gov/Version4/Release3/nctiles_daily/SIheff. Analysis was performed with Antarctic Mapping Tools for MATLAB (Greene et al., 2017b). The background image in Figs. 1, 4, and 5 is the MODIS Mosaic of Antarctica (Haran et al., 2014). Figs. 1, 4, 5, and 6 use `cmocean` colormaps (Thyng et al., 2016).

*Author contributions.* CAG conceived of this study, generated the figures, and wrote the manuscript. Analysis was conducted by CAG under the direction of DDB, with guidance from DAY. DEG and BKGF developed the ice/ocean model described in Section 4 and assisted in interpreting its results.

*Competing interests.* The authors declare that they have no conflicts of interest.

*Acknowledgements.* We thank Mason J. Fried and Denis Felikson for many helpful discussions throughout the development of this work.
This work was supported by the G. Unger Vetlesen Foundation and NSF grant PLR-1543452. Ocean modeling research was supported by the Australian Government's Cooperative Research Centre Programme through the Antarctic Climate & Ecosystems Cooperative Research Centre, the Australian Research Council's Special Research Initiative for Antarctic Gateway Partnership (Project ID SR140300001) and computing resource grants *m68*, *gh9* and *nk1* from the Australian Government National Computing Infrastructure.



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
