# Peer review of "Seasonal dynamics of Totten Ice Shelf controlled by sea ice buttressing"

_The Cryosphere, 2018_

## Referee Comment (RC1) · Anonymous Referee #1 · 30 May 2018

***Review of 'Seasonal dynamics of Totten Ice Shelf controlled by sea ice buttressing'***

**General comments:**

Greene et al. investigate velocity variations on the Totten Ice Shelf and examine the physical mechanisms that may cause these variations. Overall, the authors have succeeded in providing a denser velocity time series than previous studies and a thorough analysis of the top candidates for causing ice shelf flow variations. The paper is well written and figures are clear and appropriate. The title, abstract, and general organization are accurate, thorough, and logical.

Some revision is needed to provide consistent messaging across the paper regarding the temporal limitations of the data. The authors emphasize the importance of capturing seasonal variations, but do a somewhat poor job of acknowledging and discussing the seasonal limits of their own study. Figures 2 and 7 make this point nicely. The authors have captured more temporal coverage than previous studies, but the data still has large time gaps. There is, for example, no data to confirm that speedup begins in October – that is simply when the measurements begin in earnest and show a continued speedup from there. Similarly, I see no justification for assuming a nice seasonal sine wave as the "mean" seasonal behavior (page 5, line 10). The authors do not refer to other studies with data that might fill their time gaps or other evidence for this assumption of time evolution. Studies of seasonal velocity in Greenland outlet glaciers show a wide variety of annual patterns, including sudden slowdowns and speedups as well as more gradual changes. Why couldn't more dramatic events occur during the data gaps for Totten Ice Shelf? The authors acknowledge this on page 13, paragraph 2, but this point should be raised earlier and must be clearly reflected in the whole paper. Other instances where the overall paper obscures a point made at a specific spot in the paper are highlighted below – these need to be addressed to craft a cohesive full manuscript.

**Detailed comments (by page/line number):**

1/17. Briefly mention what the "more complex picture" is.

3/17+. It's odd to have data separated only into spring or autumn. What about summer and winter? The authors talk about all different seasons throughout the manuscript, so it becomes unclear what season is what and how the data fit into those seasons. Clarity is needed on what spring/summer/fall/winter means and what data fits into each season.

3/26. Indicate the location of the 'mid-shelf ice rumple' in Figure 1.

5/7. Why is there no overlap in the areas used for MODIS velocities v. GoLIVE velocities?

5/9+. The authors state here that 'the timing of springtime acceleration cannot be accurately determined for any given year'. Yet, language in other parts of the manuscript suggest that it can (e.g., in Figure 7 caption – '*begins* with the breakup of landfast sea ice'). The whole manuscript needs to reflect the limits of the data.

6/1-3. Here the authors use the seasonal sine wave approximation to give information about seasonal cycle amplitude, maximum, and minimum. The problems with assuming this seasonal cycle are mentioned in the general comments. Thus, it's unreasonable to give these metrics – they have little scientific or practical value.

6/6. I recommend against referencing Zwally et al. 2002. While it was the initial paper that set off the wave of research on the 'Zwally effect', it is now a poor reference for understanding the complex relationships between hydrology and glacier flow. In fact, Tedstone et al. (Tedstone, A. J., P. W. Nienow, N. Gourmelen, A. Dehecq, D. Goldberg, and E. Hanna (2015), Decadal slowdown of a land-terminating sector of the Greenland Ice Sheet despite warming, *Nature*, *526*(7575), 692–695, doi:10.1038/nature15722.), which demonstrates a long-term slowing on land-terminating areas despite increased melt, is a better reference at this point. A word of caution on the larger discussion of subglacial hydrology in the manuscript. At times (e.g., this paragraph) there is a clear distinction between the processes of subglacial hydrology that might actually influence the ice shelf v. subglacial hydrology and its influence on grounded ice (which constitute most citations in the paper). At other points, however, this point can feel muddled. Unfortunately, using the 'TIS' acronym does not help and makes it easier for the reader to forget that the study is focused on an ice shelf instead of grounded ice. As the authors go through revisions, please be conscious of keeping the fact that you are looking at *ice shelf* speeds forefront in the readers' mind.

8/19-21. This sentence is confusing and the part about the constant 300 m offset does not make sense.

10/8. Always specify '*sea* ice' if that is the subject. Check the full manuscript for this clarification.

13/last paragraph (onto next page). This paragraph discusses some specific details of the Li et al. (2016) paper without ever pulling back to the big picture of that paper to discuss this study's overall influence on interpretations of the Li et al. paper. Are the Li et al. conclusions still good ones or should the larger conclusions be reinterpreted? Also, while it's fine to point the reader to these references, try to craft this manuscript to cover all the major points so that reference to the other paper directly is not a necessity to get to the primary points regarding its (re)interpretation. The reader should come away with a sense of the pertinent conclusions of Li et al. and how they may be shifted (or not) – not only an understanding of how very specific details should be considered. This comment can be applied to any previous study the authors want to comment on.

14/14. Remove 'strength' – this paper does not include a scientific assessment of sea ice strength.

14/20. This final sentence is more declarative than I think the data supports.

14/25. Regarding 'may have aliased some previous measurements of interannual variability' – as mentioned earlier, discuss directly what these previous studies say and what the new outlook is after applying the data from this paper.

14/29+. This paragraph mixes *inter*annual basal melt and velocity changes and *intra*-annual basal melt and velocity change. I agree that the authors have done a nice job of showing how seasonal basal melt variations cannot explain seasonal speed variations, but I don't think the authors have shown that multi-year thickness changes could not play a role in multi-year speed trends.

**Typos, etc. (by line number):**

2/13. All instances of 'mélange' should have the correct accent added.

3/23. 'Throughout' is more correctly 'during' since there is no winter data to show the timing of speed changes.

**Figures:**

Figure 6. In the MODIS images it looks like the sea ice is not in contact with the glacier ice. Is there a shadow effect? Something else? Please explain/clarify.

Figure 7. Specify '*sea* ice thickness'.

---

## Referee Comment (RC2) · S. Gardner (Referee) · 15 Jun 2018

Review of: Seasonal dynamics of Totten Ice Shelf controlled by sea ice Buttressing by Greene et al.

Paper Summary: In this study the authors examine intra-annual changes in the surface velocity of the Totten Ice Shelf (TIS). Velocity measurements are acquired from feature tracking of Landsat-8 (GoLIVE, 2013-2018, 12-112 day separation) and MODIS (ImGRAFT, 2003-2017, 92-182 day separation) image pairs. Fitting a sinusoid to the MODIS velocities, by means of least squares, the authors identify a 106 m/yr fluctuation in surface velocity. From the Landsat image pairs they determine an average spring to fall speedup of 0.8 m/yr. per day. Mapped differences between spring and fall

velocities indicate that the summer speedup is concentrated towards the terminus of the ice shelf.

The authors then explore 3 likely causes for the summer speedup (surface melt, basal melt, and changes in sea ice backstress). Examining melt days determined from passive microwave data, the authors conclude the speedup precedes melt onset and therefore surface melt is unlikely to be the trigger for springtime speedup but they acknowledge that it may play a role later in the season. Through a combination of ocean modeling within the ice shelf cavity and simplified ice shelf mechanics the authors demonstrate that seasonal change in basal met rates, that have seasonal amplitudes of >8m/yr. at the grounding line and 3 m/yr. near the terminus, have little impact on rates of ice flow (several orders of magnitude below the observed signal). Lastly the authors explore changes in sea ice concentration and sea ice thickness and postulate that the breakup of fast ice in spring is the most likely trigger for the summer speedup.

Overall Opinion: The paper is well written, has a logical layout, and the analysis is transparent and easy to follow. The subject matter is appropriate for TC and will be well received by its audience. Despite the overall good quality of the manuscript I was left with a few concerns on the conclusions as drawn from the data. I see no barriers to the authors addressing these concerns in a revised manuscript.

1. My most pressing concern is the characterization of the intra-annual variability of ice shelf surface flow given the limitation in deriving surface velocities from the Landsat and MODIS images; low SNR, observations limited to polar day, and large/variable image-pair time separations. All of these conditions make it challenging to characterize intra-annual fluctuations in surface velocities. To this end I think it would be very valuable if the authors could explore the sensitivity of the least squares parameter fits to the velocity fields. For example: what is the implication of using large image-pair separations? Using bootstrapping can you better quantify the uncertainty in the fit? What does the phase and amplitude look like if you derive parameters on a pixel by pixel basis? How much do fits to the Landsat and MODIS data differ when constrained

to the period of overlap? Is a sinusoidal fit justified by the data or should the authors solely focus on the spring to fall speedup?

2. It would be very valuable if the authors could provide uncertainties with their estimates. What is the uncertainty of the estimated annual amplitude in velocity? What is the uncertainty in the modeled melt rate and respective response in modeled ice shelf velocity? What are the uncertainties in the estimated velocities and how do these propagate into the model fits (the authors could use bootstrapping to answer this)?

3. One of the 3 environmental forcings examined as a potential trigger for spring-time speedup is surface melt. Given the very low number of days that experience any liquid water at the surface, I am suspect that there is any liquid water that does not re-freeze within the first few meters of the firn column. Can the authors provide any support that this is not the case? If not I would suggest removing this section from the paper and simply state that the vast majority of meltwater will refreeze within the firn and therefore it will not impact ice shelf flow.

4. There are a few places in the manuscript, including the introduction, Section 6.3 and the conclusions, where variability in discharge and its potential aliasing in mass change estimates are presented as the motivation for this work. I don't think this is an appropriate justification. Maybe the authors could simply us the justification that improving understating of glacier mechanics/response to intra-annual changes in boundary conditions is relevant to improving glacier models and thus future projections of sea level rise.

5. The authors clearly demonstrate that seasonal changes in ice shelf thickness on the order of 0.3 to 1 m are unimportant for seasonal fluctuations in ice shelf velocity. This is well proven through their combined ocean and ice shelf modeling. They go on to conclude that changes in sea ice thickness of the same magnitude ($\sim 1 - 1.5$ m) are the cause of seasonal ice-shelf acceleration. They come to this conclusion primarily through the coincident removal of fast ice and ice shelf speedup. While I think this is

a plausible conclusion it would be helpful for the authors to discuss the mechanisms by which sea ice is able to exert such an influence. Do the authors see seasonal fluctuations in the position of the ice shelf front that could suggest a modification in the calving rate? I would think that the backstress from 1 m of sea ice would not be sufficient in itself and instead it there would need to be some mechanism by which a small force at the front of the ice shelf could disproportionately modify the fontal stress regime

---

## Author Comment (AC1) · 14 Jul 2018

We thank Alex Gardner and our anonymous reviewer for their thoughtful reading of the manuscript and for their suggestions that have led to a number of minor to critical improvements. Both reviewers noted that original manuscript was well written, logically organized, and that our analysis was accurate, thorough, and easy to follow. As such, we have made an effort to keep the original text untouched wherever possible, but not at the expense of fully addressing the reviewers' concerns.

Both reviewers stated clearly that the largest issue of concern regarded our use of a sinusoid to describe measurements of subannual velocity variability at TIS. Reviewer 1 felt there is "no justification for assuming a nice seasonal sine wave" and that providing metrics of its amplitude and phase is "unreasonable" and has "little scientific or practical value." Dr. Gardner was also skeptical due to appreciable noise in the MODIS velocity measurements used to estimate the sinusoid, and more importantly, Dr. Gardner pointed out our failure to present uncertainty estimates for the parameters of the sinusoid fit. We appreciate this feedback, as it brought to our attention the need to clarify and add analytical support for one of the key aspects of our paper.

In particular, we have followed Dr. Gardner's suggestion and added an Appendix in which we use a bootstrapping technique to estimate uncertainty in the amplitude and phase of the sinusoid fit to noisy MODIS data. The results are shown in this new figure, which now appears in the Appendix:

[Figure]

The figure shows the amplitudes and phases of sinusoids fit to 10,000 random resamplings of the MODIS velocity data. The intensity of the red color indicates data density, and distributions of each parameter are shown as histograms. Contour intervals and axis labels are shown at $1\sigma$ intervals.

The remarkably narrow skews (i.e., low uncertainty) of the amplitude and phase parameters in the figure above show that the sinusoid is robust and provides a direct response to both reviewers' concern of whether or not the sinusoid is justified by the data. To this question we state conclusively that yes, the sinusoid is justified by the data. The sinusoid is not behavior we have assumed—it is behavior we have measured.

The reviewers might be reasonably concerned that while a sinusoid fits the data well, it will not describe any higher-frequency behavior that occurs in nature. That is certainly true, but just as a linear least-squares fit provides the simplest, most robust assessment of a trend over time, a sinusoid provides the simplest, most robust assessment of cyclic behavior.

We do not attempt to model higher frequency behavior, nor would our data allow it. The MODIS data represent displacements over periods of 92 to 182 days that span the entire calendar year, which together meet Nyquist's requirement for representation of a sinusoid with a 365 day period, but do not allow higher-frequency investigation. We note here, that although we cannot detect abrupt velocity events with the MODIS data, the MODIS velocity measurements do still provide a record of all accelerations and slowdowns (albeit lowpass-filtered) by providing the sum total ice displacement between image pairs.

Our overall conclusions do not hinge on the representation of the MODIS velocity data as a sinusoid, but we do find it prudent and insightful to try to characterize the amplitude and phase of the seasonal variability, to the best that the data will allow. We have responded in further detail to the reviewers' specific comments on this point below. We have also responded to all other comments and we have incorporated most of the reviewers' suggestions, as documented below. The original reviewer comments appear in *blue italics* and our response is provided in upright black text. We believe the changes we have made in response to the reviewer feedback have resulted in a stronger, clearer, and more scientifically sound paper.

*RC1: Anonymous Referee #1:*
*General comments:*
*Greene et al. investigate velocity variations on the Totten Ice Shelf and examine the physical mechanisms that may cause these variations. Overall, the authors have succeeded in providing a denser velocity time series than previous studies and a thorough analysis of the top candidates for causing ice shelf flow variations. The paper is well written and figures are clear and appropriate. The title, abstract, and general organization are accurate, thorough, and logical.*

*Some revision is needed to provide consistent messaging across the paper regarding the temporal limitations of the data. The authors emphasize the importance of capturing seasonal variations, but do a somewhat poor job of acknowledging and discussing the seasonal limits of their own study. Figures 2 and 7 make this point nicely. The authors have captured more temporal coverage than previous studies, but the data still has large time gaps. There is, for example, no data to confirm that speedup begins in October – that is simply when the measurements begin in earnest and show a continued speedup from there. Similarly, I see no justification for assuming a nice seasonal sine wave as the "mean" seasonal behavior (page 5, line 10). The authors do not refer to other studies with data that might fill their time gaps or other evidence for this assumption of time evolution. Studies of seasonal velocity in Greenland outlet glaciers show a wide variety of annual patterns, including sudden slowdowns and speedups as well as more gradual*

*changes. Why couldn't more dramatic events occur during the data gaps for Totten Ice Shelf? The authors acknowledge this on page 13, paragraph 2, but this point should be raised earlier and must be clearly reflected in the whole paper. Other instances where the overall paper obscures a point made at a specific spot in the paper are highlighted below – these need to be addressed to craft a cohesive full manuscript.*

In response to our measurement of cyclic behavior, we point to the discussion at the top of this document, and we discuss the more detailed concerns below.

*Detailed comments (by page/line number):*
*1/17. Briefly mention what the "more complex picture" is.*

The sentence in question previously read,

*Short-term observations have identified Totten Glacier and its ice shelf (TIS) as thinning rapidly (Pritchard et al., 2009, 2012) and losing mass (Chen et al., 2009), but longer-term observations paint a more complex picture of interannual variability (Paolo et al., 2015; Li et al., 2016; Roberts et al., 2017; Greene et al., 2017a).*

Following the reviewer's suggestion we have added a brief mention of what is meant by a "more complex picture." The sentence now reads,

*Short-term observations have identified Totten Glacier and its ice shelf (TIS) as thinning rapidly (Pritchard et al., 2009, 2012) and losing mass (Chen et al., 2009), but longer-term observations paint a more complex picture of interannual variability marked by multi-year periods of ice thickening, thinning, acceleration, and slowdown (Paolo et al., 2015; Li et al., 2016; Roberts et al., 2017; Greene et al., 2017a).*

*3/17+. It's odd to have data separated only into spring or autumn. What about summer and winter?...*

The paragraph in question reads,

*To understand the spatial pattern of TIS seasonality, we developed characteristic velocity maps for spring and autumn separately...The resulting difference between spring and autumn velocities is shown in Fig. 1.*

There are myriad ways to look at these datasets, and indeed, our analytical process involved slicing the data in many different fashions that proved less insightful than the analyses we ultimately included in the manuscript. For our purposes the spring/autumn comparison hardly seems odd, as it provides a spatial overview of the difference in ice speeds between the fastest time of the year and the slowest time of the year.

*...The authors talk about all different seasons throughout the manuscript, so it becomes unclear what season is what and how the data fit into those seasons. Clarity is needed on what spring/summer/fall/winter means and what data fits into each season.*

We have stated in the text that for the GoLIVE data,

*Spring velocity was taken as the mean of 76 velocity measurements whose image pairs were obtained between June 16 and December 15. Autumn velocity was taken as the mean of 67 velocity fields from images obtained during the remainder of the year.*

Elsewhere in the text, we have been careful to discuss the results with an appropriate level of temporal precision, stating that for the GoLIVE data,

*The terminal ~50 km of TIS accelerates each year from spring to autumn, then slows during the winter.*

Given the well-constrained definition of the spring and autumn velocity data, we feel that this statement gives a physical description of the observed behavior and is unlikely to lead to any confusion. Later in the text we state that

*Everywhere beneath the ice shelf, basal melt rate reaches a maximum in autumn and a minimum in the spring.*

And we support this statement with a clearly marked figure showing the time series of basal melt rates, that the reader may inspect and should thus leave no room for confusion. The section on basal melt rates concludes,

*Thus, it is unlikely that the seasonal cycle of basal melt could explain the observed pattern of spring-to-fall acceleration of TIS.*

And again, we feel this statement is given with an appropriate level of precision.

In the Discussion section, we state that,

*In the GoLIVE dataset and in MODIS-derived velocities, we find that the outer TIS accelerates each year between spring and autumn.*

We feel that this statement is accurate and clear as it stands.

The process of scouring the manuscript for potentially misleading language helped us identify this sentence:

*We find that the outer TIS accelerates each **spring**, likely in response to lost buttressing upon the breakup of rigid sea ice at the ice shelf terminus.*

which we have modified into the following, more conservative statement:

*We find that the outer TIS accelerates each **year**, likely in response to lost buttressing upon the breakup of rigid sea ice at the ice shelf terminus.*

Other mentions of seasons include this statement:

*The seasonal variability we observe at TIS suggests that measurements acquired in the spring likely underestimate, and autumn measurements overestimate, the mean annual velocity of the ice shelf.*

which we feel is accurate and appropriate. Seasons of the year do come up a few other times in the manuscript, but only in general contexts where a higher level of precision is not warranted.

*3/26. Indicate the location of the 'mid-shelf ice rumple' in Figure 1.*

We have indicated the location of the mid-shelf ice rumple in Figure 1 by including the following text in the figure caption:

*The black line indicates the grounding line, and includes a grounded ice rumple near the center of the ice shelf (Mouginot et al., 2017b).*

*5/7. Why is there no overlap in the areas used for MODIS velocities v. GoLIVE velocities?*

There is overlap between GoLIVE and MODIS measurements in the sense that Figure 1 shows the presence of a seasonal cycle in the MODIS polygon. We realize there would be some advantages to carrying out the full analysis with overlapping measurements, both for continuity of the time series, and for measurement redundancy. However, given the different strengths and limitations of the MODIS and GoLIVE datasets, we found it beneficial to optimize the measurement areas separately for each sensor. In particular, the 15 m resolution of Landsat 8 allows us to measure a small, responsive area of the ice shelf, close to the ice shelf front, where the seasonal signal is the strongest. The 250 m MODIS pixels require a larger template chip for a unique fingerprint, and a longer time between images for reduced uncertainty in estimates of velocity. These two characteristics of MODIS mean the search area must be quite large, which prevents measurements close to the ice shelf front. In addition, the high noise of the MODIS displacement measurements means we must average over a large area to increase the signal-to-noise ratio. Thus the GoLIVE measurement area is more suited to capturing the large amplitude seasonal cycle close to the ice shelf front, whereas the MODIS dataset captures the larger-scale behavior of the ice shelf. Accordingly, these measurement areas provide two independent measures of the seasonal dynamics of TIS. We describe these points in the first paragraph of Section 2, which states,

*...Each image dataset was processed separately, using different feature tracking programs, and the resulting time series represent two independent measures of TIS velocity. The 15 m resolution of Landsat 8 permits precise displacement measurements over short time intervals, but the relatively brief four-year Landsat 8 record and limited number of cloud-free images inhibits our ability to separate interannual velocity changes*

*from seasonal variability. The MODIS record contains many cloud-free images per year from 2001 to present; however, the 250 m spatial resolution of MODIS images limits measurement precision where ice displacements are small between images. Thus, the two image datasets each offer incomplete, but complementary insights into the seasonal dynamics of TIS. Processing methods for each dataset are described below.*

*5/9+. The authors state here that 'the timing of springtime acceleration cannot be accurately determined for any given year'. Yet, language in other parts of the manuscript suggest that it can (e.g., in Figure 7 caption – 'begins with the breakup of landfast sea ice'). The whole manuscript needs to reflect the limits of the data.*

We have tried to clear up any confusion by specifying that our approach is to analyze the characteristic seasonal behavior of TIS rather than attempting to attribute particular ice shelf acceleration events to specific transient causes. To alleviate any confusion, we have clarified the section mentioned by the reviewer, which now reads,

*...subannual template matching applied to 250 m resolution MODIS images produces such noisy velocity estimates that the timing of springtime acceleration cannot be accurately determined for any given year. However, by combining data from all years we can assess the characteristic cycle of ice shelf acceleration and slowdown that occurs throughout the typical year.*

and we have edited the caption of Figure 7 to clarify that the

*characteristic springtime acceleration begins with the breakup of landfast sea ice.*

In the Discussion Section 6.2 we do discuss velocities obtained by Li et al. that were specific to the years 2009 and 2010, but we do not compare our velocity measurements to theirs for those years or any other specific years. We found no other instances in the manuscript that could imply analysis of velocities for any given year.

*6/1-3. Here the authors use the seasonal sine wave approximation to give information about seasonal cycle amplitude, maximum, and minimum. The problems with assuming this seasonal cycle are mentioned in the general comments. Thus, it's unreasonable to give these metrics – they have little scientific or practical value.*

We have not assumed sinusoidal behavior—we have measured it. And by fitting sinusoids to 10,000 random subsamplings of the measurements, we have confirmed that no matter how you slice it, there is periodicity at the 1 yr$^{-1}$ frequency, and its timing and amplitude are consistent.

To ensure that readers are not misled into thinking that we have simply assumed sinusoidal behavior, we have checked the manuscript for any instances of misleading language, and we have added the following clarifying statement the section that describes the MODIS velocity data:

*The sinusoid provides a measure of periodicity at the 1 yr⁻¹ frequency and matches observations to a root-mean-square error of 93 m yr⁻¹.*

In the General Comments section, the reviewer makes the point that "in Greenland, outlet glaciers show a wide variety of annual patterns, including sudden slowdowns and speedups as well as more gradual changes," and goes on to ask, "why couldn't more dramatic events occur during the data gaps for Totten Ice Shelf?"

There is no doubt that Totten's dynamic seasonal cycle is more complex than a simple sinusoid, but just as a linear least-squares fit can provide a first order of understanding of long-term trends (e.g., Pritchard et al., 2009, Pritchard et al., 2012 for linear trends applied to surface elevations—work that has motivated nearly a decade of Antarctic science), a sinusoidal least-squares fit provides a first-order understanding of cyclic behavior.

Regarding the potential for sudden slowdowns and speedups, such dramatic events may occur at Totten, and if they do, we have captured them. With MODIS we have measured total displacements over 92 to 182 days—In other words, we measured the displacements associated with every abrupt speedup and slowdown throughout the year, integrated over time.

Regarding the concern about data gaps throughout the year—there are none. We intentionally allowed up to 182 days temporal separation between image pairs so we could capture all ice movement that occurs at all times of the year. We could show this in Figures 3 and 7 as we do in Figure 2, with horizontal bars connecting the collection times of the image pairs, but with 565 MODIS image pairs, if each bar were a just a few pixels thick, they would clearly cover the entire calendar year with no gaps, but would blend together into an unintelligible mess. Thus, we represent each MODIS measurement as a single dot placed at its central time, but even still, the points cluster. To clear up any confusion, we have added this reminder to the caption of Figure 3:

*Velocity measurements are shown at the mean of the acquisition times of their MODIS image pairs.*

We disagree with the statement that it is unreasonable, unscientific, and impractical to fit a sinusoid to measurements of total displacement taken over periods of 92 to 182 days. The sinusoid provides a reasonable first-order understanding of the amplitude and phase of velocity variability throughout the year. Given that the large pixels of the MODIS sensor limit our temporal resolution to 92 to 182 days, Nyquist's theorem is quite clear that it would be unreasonable and unscientific to fit a higher-order model to the data. And given that we measure total displacement in each image pair, the sinusoid provides a complete assessment of velocity variability at the 1 yr⁻¹ frequency. Thus, stating the amplitude and phase of the sinusoidal behavior we measure is reasonable, scientific, practical, and meaningful.

*6/6. I recommend against referencing Zwally et al. 2002. While it was the initial paper*

*that set off the wave of research on the 'Zwally effect', it is now a poor reference for understanding the complex relationships between hydrology and glacier flow. In fact, Tedstone et al. (Tedstone, A. J., P. W. Nienow, N. Gourmelen, A. Dehecq, D. Goldberg, and E. Hanna (2015), Decadal slowdown of a land-terminating sector of the Greenland Ice Sheet despite warming, Nature, 526(7575), 692–695, doi:10.1038/nature15722.), which demonstrates a long-term slowing on land-terminating areas despite increased melt, is a better reference at this point...*

The Tedstone et al. 2015 paper is an excellent paper that found long-term glacier slowdown in the presence of increased surface melt. Their suggested mechanism for the slowdown is that more meltwater at higher elevations allows efficient drainage systems to develop farther into the ice sheet interior. Although Tedstone et al. tracked ice displacements using several hundred image pairs (and in this way is similar to our approach), they properly removed seasonal effects from their decadal trend analysis by limiting image separation times to 352 to 400 days. We seek to understand the seasonal cycle that takes place within those ~365 days, and what effects such a cycle may have on the flow of Totten. Accordingly, we describe hydrological processes that take place over the course of days to months, and we provide references accordingly. The sentence in question begins our discussion of how surface melt can affect glacier flow, and it reads,

*Surface melt has been shown to affect the flow of grounded ice in Greenland when surface water drains through moulins or crevasses to the bed, where it alters basal water pressure and allows the overlying ice to accelerate (Zwally et al., 2002; Schoof, 2010; Bartholomew et al., 2010; Andrews et al., 2014).*

The findings of Tedstone et al., 2015 do not in any way contradict this statement, and in fact their manuscript directly affirms the 'Zwally effect,' and includes a citation of Zwally, stating that "inputs of surface meltwater...lubricate the ice-bed interface, transiently speeding up the flow of ice (Zwally et al., 2002; Sole et al., 2013)." We feel it is appropriate to give credit to the originator of the idea, but we also reference some of the follow-on work that has brought a deeper understanding of the processes that are most directly related to our paper. Thus, we prefer to keep this sentence unchanged.

*...A word of caution on the larger discussion of subglacial hydrology in the manuscript. At times (e.g., this paragraph) there is a clear distinction between the processes of subglacial hydrology that might actually influence the ice shelf v. subglacial hydrology and its influence on grounded ice (which constitute most citations in the paper). At other points, however, this point can feel muddled. Unfortunately, using the 'TIS' acronym does not help and makes it easier for the reader to forget that the study is focused on an ice shelf instead of grounded ice. As the authors go through revisions, please be conscious of keeping the fact that you are looking at ice shelf speeds forefront in the readers' mind.*

This point is well taken. We have a vested interest in clarity, and we do not want to come across as careless in our language or in our treatment of the underlying physics. However, while subglacial hydrology is most closely associated with local accelerations of

grounded ice, no process occurs in isolation, and when grounded ice accelerates, it most surely influences the flow of ice downstream, though the extent to which local accelerations affect large-scale flow is largely unknown. We also leave open the possibility that surface melt can lead to shear margin weakening, which would primarily affect the flow of the floating ice shelf.

In discussions of theory, it is easy to separate these different processes and we can be quite specific. For example, the paragraph in question states clearly,

*Surface melt has been shown to affect the flow of grounded ice in Greenland when surface water drains through moulins or crevasses to the bed, where it alters basal water pressure and allows the overlying ice to accelerate (Zwally et al., 2002; Schoof, 2010; Bartholomew et al., 2010; Andrews et al., 2014). The seasonal velocity anomalies we observe at TIS are strongest near the floating ice front, so it is unlikely that the seasonal variability of TIS velocity is driven by subglacial hydrology on nearby grounded ice. However, the presence of englacial liquid water can weaken ice (Liu and Miller, 1979), and it is plausible that surface melt at TIS could percolate into the ice, weaken shear margins, and allow TIS to speed up as a result of reduced buttressing.*

When analyzing the data, we were careful to consider surface melt on grounded ice, the inner TIS, and the outer TIS separately, and we are clear about this distinction throughout the discussion. We have also been careful to separate grounded and floating ice processes in the final Discussion Section 6.1 in which we state,

*On grounded ice, seasonal velocity variability often results from surface water draining to the bed, where it can temporarily pressurize an inefficient hydrological system, allowing the overlying ice to accelerate until an efficient drainage system forms or the water otherwise evacuates (Zwally et al., 2002; Parizek and Alley, 2004; Bartholomew et al., 2010). At Totten Glacier, we detect very little seasonal velocity variability on grounded ice, and the onset of acceleration we observe on the floating ice shelf begins well before surface water is detected anywhere in the region (Fig. 7). We therefore rule out the possibility that surface melt is responsible for initiating TIS acceleration each year.*

*8/19-21. This sentence is confusing and the part about the constant 300 m offset does not make sense.*

The section in question previously read,

*Seafloor topography was based on the RTOPO dataset (Timmermann et al., 2010), while cavity geometry was inferred from ICESat-derived ice surface elevation above flotation and a constant 300 m thick offset along the central flow line. Between the central flow line and the grounding line, cavity bathymetry was linearly interpolated (see Gwyther et al., 2014, for details).*

We have changed the wording to make it more clear, replacing the two sentences above

with the following four sentences:

*Seafloor bathymetry for the deep ocean and continental shelf was taken from the RTopo-1 dataset (Timmermann et al., 2010). As RTopo-1 does not contain the cavity of TIS, we inferred the cavity geometry. Ice basal draft for the TIS cavity was obtained from ICESat-derived surface elevations, assuming hydrostatic equilibrium and a mean ice density of 905 kg m$^{-3}$ (following Fricker et al., 2001). Water column thickness was obtained by linearly interpolating from 0 m thick along the grounding line to 300 m thick along the central flow line of the ice shelf (see Gwyther et al., 2014, for details).*

*10/8. Always specify 'sea ice' if that is the subject. Check the full manuscript for this clarification.*

We have corrected this ambiguity by specifying "sea ice thickness" in all 10 instances in the manuscript that previously said only "ice thickness."

*13/last paragraph (onto next page). This paragraph discusses some specific details of the Li et al. (2016) paper without ever pulling back to the big picture of that paper to discuss this study's overall influence on interpretations of the Li et al. paper. Are the Li et al. conclusions still good ones or should the larger conclusions be reinterpreted? Also, while it's fine to point the reader to these references, try to craft this manuscript to cover all the major points so that reference to the other paper directly is not a necessity to get to the primary points regarding its (re)interpretation. The reader should come away with a sense of the pertinent conclusions of Li et al. and how they may be shifted (or not) – not only an understanding of how very specific details should be considered. This comment can be applied to any previous study the authors want to comment on.*

The original manuscript failed to provide an overview of the findings of previous TIS papers before delving into the details that are pertinent to our reinterpretation. We also failed to convey the nuance that although some of the specific velocity measurements presented by Li et al. may have been partly aliased by subannual variability, their overall findings of interannual sensitivity to ocean forcing and their grounding line flux estimates still hold. Following the reviewer's suggestion, we now begin the discussion of previous work with the following two sentences:

*Velocity variability at TIS has been investigated in three recent papers that tracked ice accelerations and slowdowns over the past few decades, and each study found that on interannual timescales, TIS dynamically responds to ocean forcing from below. We do not find any evidence that contradicts the overall findings of the previous studies, but in some cases, velocities were measured over periods of less than one year, and may have been aliased by seasonal variability...*

We then discuss some of the details of how velocity measurements were obtained and interpreted in the previous studies, and we conclude our discussion of the Li et al. results with the following two sentences:

*Despite the seasonal variability we observe near the TIS front, mass balance of an ice sheet is more meaningfully measured at the grounding line, where ice begins to have an impact on sea level. Our results show little subannual velocity variability at the grounding line, thus supporting the grounding line flux estimates by Li et al. (2016).*

*14/14. Remove 'strength' – this paper does not include a scientific assessment of sea ice strength.*

We have taken the reviewer's advice and removed the word strength. The sentence in question previously read,

*...TIS is sensitive to environmental forcing on subannual timescales, and its flow is primarily controlled by the presence and strength of sea ice at the TIS front.*

The sentence now reads,

*...TIS is sensitive to environmental forcing on subannual timescales, and its flow is primarily controlled by the presence of sea ice at the TIS front.*

*14/20. This final sentence is more declarative than I think the data supports.*

The Discussion section previously concluded as follows:

*...However, calving front processes can have far-reaching effects on glacier thickness and velocity (Nick et al., 2009), and it is possible that long-term changes in winter sea ice cover (Bracegirdle et al., 2008) could have integrated effects on TIS buttressing: The duration and thickness of sea ice cover each winter controls the total annual buttressing at the ice front, the annual flow of the ice shelf, and potentially the long-term mass balance of TIS and the Aurora Subglacial Basin.*

As this wraps up the Discussion section of the paper, we feel it is warranted to stand back and consider the implications of the processes we have reported, but following the reviewer's suggestion we have softened the language and ensured it is qualified with **can**, **possible**, **could**, **if**, and **potentially**. The section now reads:

*...However, calving front processes **can** have far-reaching effects on glacier thickness and velocity (Nick et al., 2009), and it is **possible** that long-term changes in winter sea ice cover (Bracegirdle et al., 2008) **could** have integrated effects on TIS buttressing: **If** the duration and thickness of winter sea ice control the total annual buttressing at the ice shelf front, long-term changes in sea ice cover **could** affect the annual flow of TIS, and **potentially** the mass balance of TIS and the Aurora Subglacial Basin.*

*14/25. Regarding 'may have aliased some previous measurements of interannual variability' – as mentioned earlier, discuss directly what these previous studies say and what the new outlook is after applying the data from this paper.*

Following the reviewer's previous recommendation in comment 13/last paragraph above, we have added a section that directly discusses what these previous studies say and what the new outlook is after reinterpreting the previous results with subannual variability in mind.

*14/29+. This paragraph mixes interannual basal melt and velocity changes and intra-annual basal melt and velocity change. I agree that the authors have done a nice job of showing how seasonal basal melt variations cannot explain seasonal speed variations, but I don't think the authors have shown that multi-year thickness changes could not play a role in multi-year speed trends.*

The confusion here is due to a lack of clarity on our part. The primary culprit may have been the first of the following two sentences, which previously read:

*Previous studies have investigated TIS velocity variability and have broadly concluded that interannual changes in ice velocity have been caused by sustained basal melt rate anomalies. Basal melt cannot explain the seasonal velocity variability we observe, because the seasonal amplitude of melt is too weak to produce enough thinning for an observable velocity response...*

The intent of the paragraph is to put our findings into context with previous work, but the language in the first sentence above may have inadvertently implied that we don't believe basal melt affects ice flow on interannual timescales. Hopefully this rewrite is a bit more clear:

*Previous studies have linked interannual velocity variability at TIS to periods of ice shelf thickening and thinning caused by sustained basal melt rate anomalies. On subannual timescales, however, the seasonal amplitude of basal melt variability is insufficient to produce enough thinning to elicit an observable velocity response...*

*Typos, etc. (by line number):*
*2/13. All instances of 'mélange' should have the correct accent added.*

The accent has been added for all instances.

*3/23. 'Throughout' is more correctly 'during' since there is no winter data to show the timing of speed changes.*

Agreed. 'during' is a better word choice, and the change has been made accordingly.

*Figures:*
*Figure 6. In the MODIS images it looks like the sea ice is not in contact with the glacier ice. Is there a shadow effect? Something else? Please explain/clarify.*

We have edited the caption of the figure to better describe the sea ice presence the MODIS images. The caption now states,

*Five example MODIS images (Scambos et al., 2001; updated 2018) show sea ice fastened to half of the TIS front in May and September, with dashed quadrangles indicating the region of ice concentration averaging and a gold marker denotes the location of the ECCO sea ice thickness time series.*

*Figure 7. Specify 'sea ice thickness'.*

We have edited the y axis label of Figure 7 to make the 'sea ice thickness' distinction.

*RC2: Alex Gardner:*
*Paper Summary: In this study the authors examine intra-annual changes in the surface velocity of the Totten Ice Shelf (TIS). Velocity measurements are acquired from feature tracking of Landsat-8 (GoLIVE, 2013-2018, 12-112 day separation) and MODIS (ImGRAFT, 2003-2017, 92-182 day separation) image pairs. Fitting a sinusoid to the MODIS velocities, by means of least squares, the authors identify a 106 m/yr fluctuation in surface velocity. From the Landsat image pairs they determine an average spring to fall speedup of 0.8 m/yr. per day. Mapped differences between spring and fall velocities indicate that the summer speedup is concentrated towards the terminus of the ice shelf.*

*The authors then explore 3 likely causes for the summer speedup (surface melt, basal melt, and changes in sea ice backstress). Examining melt days determined from passive microwave data, the authors conclude the speedup precedes melt onset and therefore surface melt is unlikely to be the trigger for springtime speedup but they acknowledge that it may play a role later in the season. Through a combination of ocean modeling within the ice shelf cavity and simplified ice shelf mechanics the authors demonstrate that seasonal change in basal met rates, that have seasonal amplitudes of >8m/yr. at the grounding line and 3 m/yr. near the terminus, have little impact on rates of ice flow (several orders of magnitude below the observed signal). Lastly the authors explore changes in sea ice concentration and sea ice thickness and postulate that the breakup of fast ice in spring is the most likely trigger for the summer speedup.*

*Overall Opinion: The paper is well written, has a logical layout, and the analysis is transparent and easy to follow. The subject matter is appropriate for TC and will be well received by its audience. Despite the overall good quality of the manuscript I was left with a few concerns on the conclusions as drawn from the data. I see no barriers to the authors addressing these concerns in a revised manuscript.*

We thank Dr. Gardner for his comments and suggestions. Most of the issues he raised were also raised by the anonymous reviewer, giving strong support for our need to fully address them in this revised version of the manuscript. We have addressed these concerns in response to Reviewer #1, so for conciseness we address only the remaining issues below.

*1. My most pressing concern is the characterization of the intra-annual variability of*

*ice shelf surface flow given the limitation in deriving surface velocities from the Landsat and MODIS images; low SNR, observations limited to polar day, and large/variable image-pair time separations. All of these conditions make it challenging to characterize intra-annual fluctuations in surface velocities. To this end I think it would be very valuable if the authors could explore the sensitivity of the least squares parameter fits to the velocity fields. For example: what is the implication of using large image-pair separations? Using bootstrapping can you better quantify the uncertainty in the fit? What does the phase and amplitude look like if you derive parameters on a pixel by pixel basis? How much do fits to the Landsat and MODIS data differ when constrained to the period of overlap? Is a sinusoidal fit justified by the data or should the authors solely focus on the spring to fall speedup?*

We have embraced this suggestion and added a sensitivity analysis section to the paper, in which we use bootstrapping to quantify the sensitivity of the least squares parameter fits to the velocity fields. The analysis is described in our general response at the top of this document, and in our responses to the detailed comments of Anonymous Reviewer #1. We appreciate the suggestion to use bootstrapping, because without it, our analysis may have been interpreted as arbitrary or incidental. By following the sinusoid fitting technique for 10,000 random subsamples of the data, we have shown that our measurements contain robust cyclic behavior of consistent amplitude and timing.

*2. It would be very valuable if the authors could provide uncertainties with their estimates. What is the uncertainty of the estimated annual amplitude in velocity? What is the uncertainty in the modeled melt rate and respective response in modeled ice shelf velocity? What are the uncertainties in the estimated velocities and how do these propagate into the model fits (the authors could use bootstrapping to answer this)?*

The new Appendix provides an assessment for the uncertainties in the estimates of annual amplitude in velocity. We have also included our uncertainty estimates in the main text, which now states,

*The resulting best-fit sinusoid is characterized by a 1601 m yr$^{-1}$ mean velocity, an amplitude of 106±9 myr$^{-1}$, a maximum velocity on March 21 (±1σ =5 days), and a minimum velocity on September 19 (±1σ =5 days). The sinusoid matches observations to a root-mean-square error of 93 m yr$^{-1}$. Uncertainty analysis of the sinusoid fit is explored in Appendix A.*

Quantifying uncertainty in the modeled melt rates, however, is less straightforward, as the model is forced by reanalysis data and relies on poorly constrained bathymetry as well as a number of parameterized assumptions about friction at the ice shelf base, etc. A thorough description of the model and a discussion of its uncertainties is provided in the Gwyther et al., 2014 reference we have cited in the text.

*3. One of the 3 environmental forcings examined as a potential trigger for spring-time speedup is surface melt. Given the very low number of days that experience any liquid water at the surface, I am suspect that there is any liquid water that does not re-freeze*

*within the first few meters of the firn column. Can the authors provide any support that this is not the case? If not I would suggest removing this section from the paper and simply state that the vast majority of meltwater will refreeze within the firn and therefore it will not impact ice shelf flow.*

In Greenland and in mountain glaciers around the world, surface meltwater can be sufficiently abundant to drain fully to the bed and lead to ice acceleration. Surface melt has not previously been explored at Totten, but we show that it does not play a major role in ice dynamics here, rather than simply assuming it. This finding in itself may be meaningful to anyone wanting to understand what does or does not affect the flow of Totten.

We also note that bed lubrication/pressurization is not the only process by which surface melt can affect ice speed. In Greenland, it has been shown that surface melt must only make its way into crevasses, where it can weaken shear margins without reaching the bed, and lead to ice acceleration. We show that this process is not a primary contributor to seasonal variability at Totten, and again, we feel that this brings meaningful understanding to the dynamics of Totten.

Given the major role that surface meltwater plays in the seasonal variability of glacier dynamics elsewhere in the world, we feel that it is important to report our findings about the role of surface melt for Totten.

*4. There are a few places in the manuscript, including the introduction, Section 6.3 and the conclusions, where variability in discharge and its potential aliasing in mass change estimates are presented as the motivation for this work. I don't think this is an appropriate justification. Maybe the authors could simply us the justification that improving understating of glacier mechanics/response to intra-annual changes in boundary conditions is relevant to improving glacier models and thus future projections of sea level rise.*

We do mention aliasing in the introduction, stating that

*The current best estimates of Totten Glacier and TIS mass budgets have been calculated using a mosaic of surface velocity measurements collected at different times throughout the year (Rignot et al., 2013); however, such estimates have been built on an unconfirmed assumption that ice velocity does not vary on subannual timescales. Where glacier flow varies throughout the year, it is possible that velocity measurements collected over short time intervals may lead to inaccurate estimates of annual mass balance or incorrect interpretation of interannual changes in velocity. Furthermore, most common methods of ice velocity measurement, such as satellite image feature tracking or in-situ GPS measurements taken over the course of a field season, are strongly biased toward summer acquisition and may not accurately represent winter ice dynamics. Wherever seasonal velocity variability exists, it is important to consider how ice velocity is measured, and how the measurements can be interpreted.*

Aliasing is not mentioned in Section 6.3, but we do bring it up again in the Conclusions section, stating,

*We find that TIS has a characteristic seasonal velocity profile, which could lead to inaccurate estimates of the annual mass balance of TIS, and may have aliased some previous measurements of interannual variability. Annual ice velocity maps are now available covering most of Antarctica (Mouginot et al., 2017c), but interpreting such datasets at TIS and elsewhere requires understanding where ice velocity varies seasonally and by how much. Our results provide context for how and where such velocity mosaics may be used to interpret interannual change at Totten Glacier.*

We feel that this is an important discussion because the concept of aliasing has previously been neglected or worked around in nearly every study of long-term velocity change in Antarctica. We show that the natural seasonal cycle of ice dynamics at Totten is on the same order as variability that has previously been attributed to long-term change. Identifying and separating intra-annual variability from interannual variability is of critical importance for interpreting and understanding the causes of dynamic change at Totten and elsewhere around Antarctica.

*5. The authors clearly demonstrate that seasonal changes in ice shelf thickness on the order of 0.3 to 1 m are unimportant for seasonal fluctuations in ice shelf velocity. This is well proven through their combined ocean and ice shelf modeling. They go on to conclude that changes in sea ice thickness of the same magnitude (1 – 1.5 m) are the cause of seasonal ice-shelf acceleration. They come to this conclusion primarily through the coincident removal of fast ice and ice shelf speedup. While I think this is a plausible conclusion it would be helpful for the authors to discuss the mechanisms by which sea ice is able to exert such an influence. Do the authors see seasonal fluctuations in the position of the ice shelf front that could suggest a modification in the calving rate? I would think that the backstress from 1 m of sea ice would not be sufficient in itself and instead it there would need to be some mechanism by which a small force at the front of the ice shelf could disproportionately modify the fontal stress regime*

The point about the influence of ~1 m of ice at the ice shelf base versus the same ice thickness at the ice shelf front is well taken. But as in architecture, the placement of structural supports is critical.

In the Introduction and in the Discussion section 6.1 we go into significant depth, describing the array of different studies that have shown how the presence of sea ice or ice melange can temporarily prevent calving, inhibit crevassing near the ice shelf front, or maintain the structural integrity of the ice shelf by preventing calved icebergs from rotating away from the ice front. Temporary reductions in calving can preserve internal stresses in the ice shelf and slow the flow of the ice. The modeling studies we describe and reference have investigated these processes in much more detail than we can consider given the limitations of our observational data.

The calving front position would indeed be an insightful time series for this analysis, but due to the large megaripples at Totten, digitizing the location of the ice front can lead to tremendous uncertainty based on visual interpretation of ice features. For example, this is a Landsat 8 image of the Totten Ice Shelf front:

[Figure]

The image above is 40 km wide, and when tasked with identifying the structural bounds of the ice shelf, we find that we cannot confidently distinguish between intact shelf ice, sea-ice-fastened icebergs, and "loose teeth" that may be partly connected to the ice shelf without supporting the full stress regime of the ice shelf. Uncertainties in identifying the structural bounds of the ice shelf would be on the order of kilometers or more, and would ultimately lead to an analysis of the interpretation of the ice shelf front location, rather than interpretation of a physically meaningful time series. Given the spatial and temporal limitations of all data in this region, here we can only observe the end members of the process, and use what is known from modeling studies and observations elsewhere in the world to infer the small-scale processes that are occurring in between. Accordingly, we do not attempt to directly measure the calving rate of the ice shelf in this revised manuscript.

---

## Author Response (AR2)

Dear Dr. Matsuoka,

Thank you for reviewing our revised manuscript. We have adopted most of your suggested changes, as we describe below. Please let us know if you have any questions or concerns.

The revised version of the manuscript and the response letter is clear and well articulates the arguments that the authors have made. The both reviewers concerned the validity of sinusoidal fitting. The authors followed the suggestion of Dr. Gartner to use a bootstrapping technique and demonstrated that the best estimate of the sinusoidal approximation is robust. Also, the authors clarify in the response letter and the manuscript that it is likely that ice-flow variations are more complicated than the simple sinusoidal curve can capture. I am happy to see that these points are well presented. However, in my opinion, the authors do not adequately exclude a possibility that there is no seasonal variation of the flow speed at all. Thus, I would like to propose the authors to perform p tests with a null hypothesis that there is no seasonal variation of the flow speed. If this hypothesis is rejected with a very high confidence, it further supports the argument...

We have taken this suggestion and we now mention the correlation coefficient between observations and sinusoid fit along with the statistical significance of the correlation in the main text. The final paragraph of Section 2.2 now states,

*...The resulting best-fit sinusoid is characterized by a 1601 m yr$^{-1}$ mean velocity, an amplitude of 106 +/- 9 m yr$^{-1}$, a maximum velocity on March 21 (1σ = 5 days), and a minimum velocity on September 19 (1σ = 5 days). The sinusoid provides a measure of periodicity at the 1 yr$^{-1}$ frequency and matches observations to r = 0.472 (p = 6x10$^{-33}$). A complete description of the sinusoid fit and a full uncertainty analysis are provided in Appendix A.*

We have also added the following paragraph to the beginning of the uncertainty analysis section:

*To verify the presence of seasonal variation in ice flow speed, we performed a p test using the null hypothesis that the amplitude of seasonal variability is 0 m yr$^{-1}$. The 565 MODIS velocity measurements match the sinusoid fit by least squares with a Pearson correlation coefficient of r = 0.472 and a corresponding $t_{statistic}$ = 12.70. The probability of the null hypothesis that there is no seasonal cycle is p = 6x10$^{-33}$, and thus we reject it in favor of the alternate hypothesis that cyclic seasonal behavior is present at TIS.*

...In addition, I would request authors to include error estimates of individual flow-velocity measurements in Sections 2.1 and 2.2.

The short answer is that error estimates for individual flow velocity measurements are not directly available from template matching algorithms. For this reason, we deliberately designed our analysis to ensure that we do not to analyze individual

velocity measurements at any point in this manuscript. Our analyses and conclusions instead lean on the climatological sinusoid fit to the measurements. Error estimates of the sinusoid we analyze are quantified by bootstrap analysis, which gives a total estimate of error in the underlying measurements and the mismatch between true behavior and the sinusoidal approximation.

A brief description of the difficulty of obtaining error estimates for individual measurements follows.

In theory, template matching should be accurate to about a quarter of a pixel displacement, which is easy enough to convert to velocity uncertainty by dividing by the *dt* between images. However, ice deformation, migrating sastrugi, snow accumulation, or partial cloud cover can complicate things quite a bit.

Part of the problem is that the algorithm might accurately track migrating sastrugi, whereas we are interested in the motion of the underlying ice. Quantifying the uncertainty in the difference between sastrugi migration and ice motion is not straightforward. In most cases, sastrugi is probably not the culprit of erroneous displacement measurements. Rather, uncertainty can result from indistinct peaks in the correlation or a high correlation peak that is surrounded by other highly correlated values. This figure from Scambos et al., 1992 depicts the concept of template matching:

[Figure]

The reference chip from the first image is chosen, and it is compared to the entire search area of the second image. Wherever the correlation index is the highest between the reference chip and the second image, that is assumed to correspond to the ice displacement. However, the cartoon above shows an ideal case where the correlation peak is quite distinct from its surroundings. Quite often, correlation values may be high, but the curve is nearly flat. Other times, correlations everywhere within the search area may be quite low, yet a small peak can provide

an accurate measure of displacement. Other times still, multiple correlation peaks can be seen within a given search area. Which peak is the correct one? How do you distinguish between a true peak versus a few pixels of noise? How broad of a correlation peak is too broad?

Because it is unclear what level of uncertainty results from these thresholding decisions in template matching, different data providers have gone about trying to quantify uncertainty by different means. The MEaSUREs annual velocity dataset (https://nsidc.org/data/nsidc-0720) averages several short-term displacement measurements collected each year, and presents two measures of uncertainty. They do provide estimated errors in the x and y directions, while cautioning that "*these values should be used more as an indication of relative quality rather than absolute error.*" To give more insights into potential error, they also provide the standard deviations of x and y displacement and the total number of displacement measurements contributing to the average velocity calculated for each pixel. In other words, because the MEaSUREs annual velocity dataset is unable to provide uncertainty estimates for individual velocity measurements, they present only the statistics of the velocity measurements they obtain for each year. In our present manuscript, we are looking at how ice velocity varies within each year, so such values are useless.

The GoLIVE dataset (https://nsidc.org/data/NSIDC-0710/versions/1) we use provides individual displacement measurements, so unlike MEaSUREs, it cannot provide statistical measures of uncertainty. Thus, for each measurement, for each pixel, GoLIVE provides the value of the peak correlation, quantitative measures of the curvature of the correlation curves in the x and y directions which may be used to determine how sharp the correlation index peak is in each direction, and the value of the second-largest peak in correlation. GoLIVE then leaves it up to the users to decide, given these metrics, what type of quality threshold may be appropriate for any given work. No other metric of error is provided in the GoLIVE dataset.

Our MODIS velocity measurements were generated with a template matching algorithm that is similar to the one used by GoLIVE, so our quality metrics are similar to theirs and we have no direct measure of uncertainty for any individual measurements.

For the reasons described above, template matching is characteristically noisy, which we have dealt with by averaging the GoLIVE and MODIS velocity measurements over large regions of the ice shelf, with the idea that noise tends to cancel itself while the signal remains strong (as in Greene et al., 2017 and Greene & Blankenship, 2018). This averaging approach might suggest that we could get an idea of uncertainty by assessing the statistics of the displacement measurements within the averaging region for any given image pair. However, that approach would assume that errors from one pixel to the next are uncorrelated, which is almost certainly untrue. While some uncorrelated noise varies from pixel to pixel, reference chips and search areas partially overlap, clouds or other physical effects can

influence arbitrary regions of the averaging area, and geolocation errors in the underlying images affect entire measurements to an unknown degree. Without a clear way to assess the spatial correlation of errors, we cannot responsibly quantify the uncertainty for individual measurements. Thus, we analyze only the climatological sinusoid, whose uncertainties are well constrained.

The authors presented the uncertainty quantification in Appendix A to keep the manuscript structure as close as possible to the original form, which was well received by the reviewers. However, depending on the outcome of the proposed tests above, please consider reorganizing the manuscript and present these uncertainty quantifications together at the end of Section 2.2.

In the main text we show the presence of seasonal variability in two independent datasets, in two separate locations on the ice shelf. In the Appendix, the bootstrapping analysis we implemented in response to Dr. Gardner's suggestion effectively shows that the chance of no seasonal variability (amplitude 0 m yr$^{-1}$) lies 12-$\sigma$ below the observed amplitude (106 m yr$^{-1}$) of the least squares fit. We also describe a $p$ test in which we show that the possibility of no seasonal variability can be rejected at the significance level of $p = 6 \times 10^{-33}$. We feel that the multiple lines of evidence we present in the main text and in the Appendix are more than sufficient to convince readers that seasonal variability exists at Totten, and inserting multiple paragraphs into the main text as proof would be an unnecessary distraction from the physics at hand. We have contained most of the uncertainty analysis to the Appendix in this revision, but we are happy to move it into the main text if the editor feels it would belong better there.

Below, you find some very minor suggestions for improvement.

- Fig. 1: the calving front is also indicated by black, so revise the caption accordingly....

We have revised the caption, which now reads,

*The calving front and grounding line are shown in black, including a grounded ice rumple near the center of the ice shelf.*

...Also, revise the second sentence in the caption to "Towards the ice calving front right bottom of the image..." so the shape and ice flow direction of TIS is clear for all readers...

We have changed the text to the suggested wording.

...The label for the colorbar should be revised to ".... (m yr-1)" from "(m/a)"...

We have changed the colorbar label as requested.

...Strictly speaking, velocity is a vector and speed is a scholar. I feel that the authors tend to describe both velocity and speed as velocity, which is acceptable for me. However, then please keep the terminology simple and uniform; consider changing "speed" in the colorbar label and elsewhere to velocity or at least check the clarity of the entire text one more time.

We, too, lament this unfortunate convention in the glaciological community. We have nonetheless opted for convention rather than correctness, and we have changed the colorbar label to 'velocity' as suggested.

- Fig. 2: Unit of the ordinate should be "m yr-1", not "m/yr". And consider plotting the speed in the unit of km/a (to be more compatible with Figs. 3 and 7).

We have implemented both changes as requested.

- P6L6-7 and P8L10-11: do you indicate the plus/minus one sigma (i.e. 2 sigma) or only one sigma? Remove the plus/minus signs from the sigma, or add this sign to the right hand side of the embedded equation as well.

Ah, yes, we understand the ambiguity now that it has been pointed out. We have followed the suggestion by removing the +/- symbol.

- Fig. 5 caption: As far as I can read from the figure, I am not convinced that melt anomalies propagate in a clockwise fashion around the cavity. If this point is not important then please delete it. If important, revise the figure or explain more clearly.

This section focuses on the magnitude, timing, and spatial distribution of basal melt at TIS, and Figure 5 shows all three. The phase of the melt rate is at the heart of our evidence for eliminating basal melt from the list of potential causes of the seasonal velocity variability we observe. The direction of circulation is not integral to our conclusions per se, but we nonetheless describe the circulation in a few brief words to provide context for understanding how the melt rate signal propagates around the ice shelf cavity. The wording is brief enough that uninterested readers can skip by it without losing the main story, but descriptive enough that curious readers can explore the figure and see the pattern themselves.

Melt rate anomalies tend to initiate at the lower left hand corner of the ice shelf, where the figure shows a zero-day phase anomaly. Following the grounding line clockwise up the left side of the ice shelf, we pass the 10 day lag contour, then the 20 day lag contour, then the 30 day lag contour. Longer lag times extend from the deepest part of the grounding line toward the ice shelf front along the right side of the figure because anomalies reach the "three o'clock" position of the ice shelf only after passing the "nine o'clock" position. Near the circulation's exit on the lower right hand side where the ice shelf front abuts Law Dome, we see a 40 day contour, which is the longest lag contour and completes the circulation regime.

The pattern described above is shown in Fig 5, and it describes what we have observed in poring over decades of high-resolution melt rate data. It is also in keeping with the reports of clockwise circulation described in the Gwyther et al. 2014 reference we cite. Of course, the model attempts to replicate the complexities of reality, so the contour map we present in Fig 5 is by no means an idealized picture of the circulation. Nonetheless, the contour map presents the phase of the melt rate as elegantly as we know how, and the physical process it depicts is described in the caption, which reads,

*...Gray contours show melt rate lag times in days relative to anomalies at the ice front, indicating melt anomalies propagate in a clockwise fashion around the cavity...*

We believe that this statement is true, it is meaningful to anyone who wonders if the nonuniform timing of melt rate anomalies throughout the cavity might affect our results, and it is brief enough that readers can understand it and move on without distraction. We feel that figure captions should do more than simply identify the markers that are used in a figure, and should instead be used to help viewers understand the physical processes and important relationships depicted in the figure. We believe the wording above accomplishes this goal, and we prefer to keep the caption as it is.

- Spell out ECCO as "Estimating the Circulation and Climate of the Ocean" at least in the reference list of Fukumori et al.

We have included the full definition of ECCO where it first appears in the section describing the basal melt rate model.

- Fig. 6: add goldern ECCO location marker in the two most right satellite images taken in April and May 2016.

We have clarified the reason for the missing golden markers by stating in the figure caption that they are *
[revised manuscript text omitted]